# The firn meltwater Retention Model Intercomparison Project (RetMIP): Evaluation of nine firn models at four weather station sites on the Greenland ice sheet

Baptiste Vandecrux[1,2], Ruth Mottram[3], Peter L. Langen[3], Robert S. Fausto[1], Martin Olesen[3], C. Max Stevens[4], Vincent Verjans[5], Amber Leeson[5], Stefan Ligtenberg[6], Peter Kuipers Munneke[6], Sergey Marchenko[7], Ward van Pelt[7], Colin Meyer[8], Sebastian B. Simonsen[9], Achim Heilig[10], Samira Samimi[11], Shawn Marshall[11], Horst Machguth[12], Michael MacFerrin[13], Masashi Niwano[14], Olivia Miller[15], Clifford I. Voss[16], Jason E. Box[1]

[1] Geological Survey of Denmark and Greenland, Copenhagen, Denmark.
[2] Department of Civil Engineering, Technical University of Denmark, Lyngby, Denmark.
[3] Danish Meteorological Institute, Copenhagen, Denmark
[4] Department of Earth and Space Sciences, University of Washington, WA USA
[5] Lancaster Environment Centre, Lancaster University, Lancaster, UK
[6] IMAU, Utrecht University, The Netherlands
[7] Department of Earth Sciences, Uppsala University, Uppsala, Sweden
[8] Thayer School of Engineering, Dartmouth College
[9] National Space Institute, Technical University of Denmark, Kgs. Lyngby, Denmark
[10] Department of Earth and Environmental Sciences, LMU, Munich, Germany
[11] Department of Geography, University of Calgary, Calgary, AB, Canada
[12] Department of Geosciences, University of Fribourg, Switzerland
[13] Cooperative Institute for Research in Environmental Sciences, University of Colorado, Boulder, CO, USA
[14] Meteorological Research Institute, Japan Meteorological Agency, Tsukuba, 305-0052 Japan
[15] U. S. Geological Survey, Utah Water Science Center, Salt Lake City, UT, USA
[16] U. S. Geological Survey, Menlo Park, CA, USA

*Correspondence to*: B. Vandecrux (bav@geus.dk)

**Abstract.** Perennial snow, or firn, covers 80% of the Greenland ice sheet and has the capacity to retain surface meltwater, influencing the ice sheet mass balance and contribution to sea level rise. Multi-layer firn models are traditionally used to simulate firn processes and estimate meltwater retention. We present, intercompare and evaluate outputs from nine firn models at four sites that represent the ice sheet's dry snow, percolation, ice slab and firn aquifer areas. The models are forced by mass and energy fluxes derived from automatic weather stations and compared to firn density, temperature and meltwater percolation depth observations. Models agree relatively well at the dry snow site while, elsewhere, their meltwater infiltration schemes lead to marked differences in simulated firn characteristics. Models accounting for deep meltwater percolation overestimate percolation depth and firn temperature at the percolation and ice slab sites but accurately simulate recharge of the firn aquifer. Models using Darcy's law and bucket schemes compare favorably to observed firn temperature and

meltwater percolation depth at the percolation site, but only the Darcy models accurately simulate firn temperature and percolation at the ice slab site. Despite good performance at certain locations, no single model currently simulates meltwater infiltration adequately at all sites. The model spread in estimated meltwater retention and runoff increases with increasing meltwater input. The highest runoff was calculated at the KAN_U site in 2012 when average total runoff across models ($\pm 2\sigma$) was $353 \pm 610$ mm w.e., about $27 \pm 48$ % of the surface meltwater input. We identify potential causes for the model spread and the mismatch with observations and provide recommendations for future model development and firn investigation.

## 1. Introduction

Responding to higher air temperatures and increased surface melt, the Greenland ice sheet has been losing mass at an accelerating rate over recent decades and is responsible for about 20% of observed global sea level rise (Van den Broeke et al., 2016, IMBIE Team, 2020). Increasing temperatures have introduced melt at higher elevations where it was previously seldom observed (Nghiem et al., 2012). In these colder, elevated areas, snow builds up into a thick layer of firn. Increased surface melt in the firn area of the Greenland ice sheet affects the firn structure (Machguth et al., 2016; Mikkelsen et al., 2016), density (de la Peña et al., 2015; Vandecrux et al., 2018), air content (van Angelen et al., 2013; Vandecrux et al., 2019) and temperature (Polashenski et al., 2014; Van den Broeke et al., 2016). These changing characteristics impact the firn's meltwater storage capacity; either through its ability to refreeze meltwater (Pfeffer et al., 1991; Braithwaite et al., 1994; Harper et al., 2012) or to retain liquid water in perennial firn aquifers (e.g. Forster et al., 2014; Miège et al., 2016). Meltwater refreezing can for instance form continuous ice layers that are several meters thick (MacFerrin et al., 2019). These ice slabs impede vertical meltwater percolation, enhance surface-water runoff (Machguth et al., 2016; Mikkelsen et al., 2016; MacFerrin et al., 2019) and lower the surface albedo (Charalampidis et al., 2015), further amplifying Greenland's contribution to sea-level rise. The evolution of firn on the Greenland ice sheet is important for two additional reasons: first, knowledge about how firn air content evolves through time is necessary for the conversion of space-borne observations of ice-sheet volume change into mass change (e.g. Sørensen et al., 2011; Zwally et al., 2011). Secondly, the depth of firn to ice transition, as well as the mobility of gases through the firn before they are trapped in bubbles within glacial ice, are necessary for the interpretation of ice cores and heavily depend on the fine coupling between the firn characteristics and surface conditions (e.g. Schwander et al., 1993).

Snow and firn models have been traditionally used to calculate the evolution of firn characteristics and meltwater retention at scales ranging from tens of meters to tens of kilometers. The performance of these models, when coupled to regional and global climate models, has a direct impact on the fidelity of ice-sheet mass-balance calculations (Fettweis et al., 2020) and sea-level change estimations (Nowicki et al., 2016). In previous work, Reijmer at al. (2012) suggested that, provided reasonable tuning, simple parameterizations of the subsurface processes calculate refreezing rates for the Greenland ice sheet in agreement with results from physically based, multilayer firn models. However, spatial patterns varied widely and evaluation against field observations remained challenging. Steger et al. (2017) and more recently Verjans et al. (2019)

investigated the impact of meltwater infiltration schemes on the simulated properties of the firn in Greenland. These studies highlighted the potential of deep-percolation schemes, for instance for the simulation of firn aquifers, but also the sensitivity of simulated infiltration to the firn structure and hydraulic properties. In these previous studies, the surface conditions were prescribed by a regional climate model. Inaccuracies in this forcing could therefore explain some of the deviation between model outputs and firn observations and prevented a full assessment of different firn model designs.

The meltwater Retention Model Intercomparison Project (RetMIP) compares results from nine firn models currently used for the Greenland ice sheet. The models are forced with consistent surface inputs of mass and energy and simulations are performed at four sites where surface conditions could be derived from automatic weather station (AWS) observations and where firn observations are available. These four sites were chosen to represent various climatic zones of the Greenland ice sheet firn area: the dry snow area, where melt is rare and temperatures are low, is represented by Summit; the percolation area, where melt occurs every summer at the surface, infiltrates in the snow and firn and refreezes there, is represented by Dye-2; ice slab regions, where a thick ice layer hinders deep meltwater percolation, is represented by KAN_U; and firn aquifer regions, where infiltrated meltwater remains liquid at depth is represented by FA. At each site, we compare simulated temperature, density and the resulting meltwater infiltration patterns between models and to in situ measurements. We discuss model features that can be responsible for model spread and deviation from observations. Lastly, we evaluate how differences in simulated firn characteristics result in various simulated refreezing and runoff values at sites where melt and/or runoff occur and attempt to quantify uncertainties linked to firn models.

## 2. Models

The multi-layer firn models investigated here are listed in Table 1. They all have density, temperature, and liquid water content as prognostic variables and apply a framework whereby firn is divided into multiple layers for which these characteristics can be calculated. The number of layers varies in each model (Table 2) and we distinguish between two distinct types of layer management strategies: all models except DMIHH and MeyerHewitt follow a Lagrangian framework, i.e. they add new layers at the top of the model column during snowfall and these layers are advected downward as new material accumulates at the surface. DMIHH and MeyerHewitt follow an Eulerian framework in which the layers have either fixed mass or fixed volumes. During snowfall, new material is added to the first layer and an equivalent mass/volume is transferred by each layer to its underlying neighbor. At each time step, the models calculate firn density according to different densification formulations and update the layer temperature using different values of thermal conductivity (Table 2). The DMIHH, GEUS and DTU models have a fixed temperature at the bottom of their column (Dirichlet boundary condition) while other models have a fixed temperature gradient (Neuman boundary condition).

All models simulate meltwater percolation and transfer water vertically from one layer to the next according to the routines listed in Table 2. They also simulate meltwater refreezing and latent heat release. All models simulate the retention of

meltwater within a layer due to capillary suction, either explicitly (MeyerHewitt and CFM model) or, for all the other models, parameterized through the use of an irreducible water content (Coléou and Lesaffre, 1998; Schneider and Jansson, 2004). When meltwater cannot be transferred to the next layer or be retained within the layer by capillary suction, lateral runoff can occur according to model-specific rules (Table 2). The background and specifics of each model are described in greater detail in the following paragraphs.

**Table 1: Models evaluated in this study.**

| Model code name | Developing institute | References |
| --- | --- | --- |
| CFM-Cr<br>CFM-KM | University of Washington, Lancaster University | Stevens et al. (2020),<br>Verjans et al. (2019) |
| DTU | Technical University of Denmark – National Space Institute | Sørensen et al. (2011), Simonsen et al. (2013) |
| DMIHH | Danish Meteorological Institute | Langen et al. (2017) |
| GEUS | Geological Survey of Denmark and Greenland | Vandecrux et al. (2018, 2020a) |
| IMAU-FDM | Institute for Marine and Atmospheric research Utrecht (IMAU), Utrecht University | Ligtenberg et al. (2011, 2018),<br>Kuipers Munneke et al. (2015) |
| MeyerHewitt | Thayer School of Engineering, Dartmouth College | Meyer and Hewitt (2017) |
| UppsalaUniBucket<br>UppsalaUniDeepPerc | Uppsala University | Van Pelt et al. (2012, 2019),<br>Marchenko et al. (2017) |

### 2.1. CFM-Cr and CFM-KM models

The Community Firn Model (CFM) is an open-source, modular model framework designed to simulate a range of physical processes in firn (Stevens et al., 2020). The number of layers for a particular model run is fixed and determined by the accumulation rate and time-step size. New snow accumulation at each time step is added as a new layer, and a layer is removed from the bottom of the model domain. A layer-merging routine prevents the number of layers from becoming too large. CFM-Cr and CFM-KM use the Crocus (Vionnet et al., 2012) and Kuipers Munneke et al. (2015) densification schemes,

respectively (Table 2). Both use the same meltwater percolation scheme: a dual-domain approach that closely follows the implementation of the SNOWPACK snow model (Wever et al., 2016). It accounts for the duality of water flow in firn by simulating both slow matrix flow and fast, localized, preferential flow (Verjans et al., 2019). In the matrix flow domain, water percolation is prescribed by the Richards Equation; ice layers are impermeable, and runoff is allowed. In contrast, water in the preferential flow domain can bypass such barriers and no runoff is simulated. Water is exchanged between both domains as a function of the firn layer properties: density, temperature and grain size. As such, when water in the matrix flow domain accumulates above an ice layer, it is progressively depleted by runoff and by transfer of water into the preferential flow domain. In the deepest firn layers, above the impermeable ice-sheet, water accumulates, and no runoff is prescribed, which allows for the build-up of firn aquifers.

## 2.2. DTU model

The DTU firn model was developed to derive the Greenland ice sheet mass balance from the satellite observations of ice sheet elevation change (Sørensen et al., 2011) and to describe the firn stratigraphy and annual layers in the dry-snow zone along the EGIG-line in central Greenland (Simonsen et al., 2013). The DTU model uses the densification scheme from Arthern et al. (2010) and a bucket scheme for meltwater infiltration and retention. If meltwater is conveyed to a model layer, the water is refrozen if sufficient pore space and cold content are available in the layer. Additional liquid water can be retained in a layer by capillary forces calculated after Schneider and Jansson (2004). This formulation does not allow for the formation of firn aquifers. Percolation continues until the water encounters a layer at ice density or the bottom of the model where, in both cases, it is assumed to run off. The model follows a Lagrangian scheme of advection of layers down into the firn and the model layering is defined by the time-stepping of the model.

## 2.3. DMIHH model

The DMIHH model was developed to provide firn subsurface details for the HIRHAM regional climate model experiments (Langen et al., 2017). DMIHH employs 32 layers within which snow, ice and liquid water fractions can vary and where each layer has a constant mass. Layer thicknesses increase with depth to increase resolution near the surface and give a full model depth of 60 m water equivalent (w.e.). Mass added at the surface (e.g., snowfall) or removed as runoff causes the scheme to advect mass downward or upward to ensure the constant w.e. layer thicknesses. DMIHH uses Darcy's law to describe meltwater infiltration. In addition to the saturated and unsaturated hydraulic conductivities (Table 2), the water flow through layers containing ice follows the model of Colbeck (1975) for a snowpack with discontinuous ice layers. A parameter describing the ratio between the characteristic distance between two adjacent ice lenses and the characteristic width of an ice lens was set to 1, meaning that ice lenses have a horizontal extent of half the unit area. A layer is considered impermeable if its bulk dry density exceeds 810 kg m$^{-3}$. Runoff is calculated from the water in excess of the irreducible saturation with a characteristic local runoff time-scale that increases as the surface slope tends to zero (Zuo and Oerlemans, 1996), with the

coefficients of the time-scale parameterization from Lefebre et al. (2003). DMIHH has an initial value of 0.1 mm for the grain diameter of freshly fallen snow. The column grain size distribution is initialized in these experiments as columns taken at the specific sites from the spinup experiments performed by Langen et al. (2017).

## 2.4. GEUS model

The GEUS model is based on the DMIHH model (Langen et al., 2017) and is further developed in Vandecrux et al. (2018, 2020a). The main differences from DMIHH are the Lagrangian management of model layers and the increased vertical resolution with 200 layers. As in the DMIHH model, the layer's ice content decreases its hydraulic conductivity according to Colbeck (1975) but the ice layer geometry parameter was set to 0.1 as detailed in Vandecrux et al. (2018). Water exceeding the irreducible water content that could not percolate downward is available for runoff and is removed from the layer at a rate that depends on the firn characteristics and on surface slope, according to Darcy's law. More details about this runoff scheme are provided in the Supplementary text S1.

## 2.5. IMAU-FDM

The IMAU-FDM model has been used in combination with the RACMO regional climate model in Greenland, Arctic Canadian ice caps, and Antarctica. Firn compaction follows a semi-empirical, temperature-based equation from Arthern (2010). The compaction rate is tuned to observations from Greenland firn cores using an accumulation-based correction factor (Kuipers Munneke et al., 2015). IMAU-FDM includes meltwater percolation following a tipping-bucket approach. Percolating meltwater is refrozen if there is space available in the layer, and if the latent heat of refreezing can be released in the layer. As opposed to other models in this study, runoff is not allowed over ice layers, but only when percolating meltwater has reached the pore close-off depth. Upon reaching that depth, runoff is instantaneous. The rationale for allowing percolation through thick ice slabs is that IMAU-FDM is mainly used to simulate firn at scales of tens to hundreds of square kilometers, and at these spatial scales, meltwater is assumed to always find a way through even the thickest of ice slabs.

## 2.6. MeyerHewitt model

Meyer and Hewitt (2017) present a continuum model for meltwater percolation in compacting snow and firn. The MeyerHewitt model includes heat conduction, meltwater percolation and refreezing, as well as mechanical compaction using the empirical Herron and Langway (1980) model. In the MeyerHewitt model, water percolation is described using Darcy's law, allowing for both partially and fully saturated pore space. Water is allowed to run off from the surface if the snow is fully saturated. Using an enthalpy formulation for the problem, the MeyerHewitt model is discretized using an Eulerian, conservative finite volume method that is fixed to the surface.

**Table 2: Model characteristics.**

| Model | Discretization | Meltwater infiltration | Hydraulic conductivity (Saturated, unsaturated) | Firn densification | Runoff calculation | Thermal conductivity |
|---|---|---|---|---|---|---|
| CFM-Cr | Unlimited number of layers, Lagrangian | Richards equation and dual-domain preferential flow scheme (Wever et al., 2016; Verjans et al., 2019) | Calonne et al. (2012); van Genuchten (1980) with coefficients from Yamaguchi et al. (2012) | Vionnet et al. (2012) | Zuo and Oerlemans (1996) | Anderson (1976) |
| CFM-KM | | | | Kuipers Munneke et al. (2015) | | |
| DTU | Dynamically allocated, based on accumulation rates, timestep and depth range, Lagrangian | Bucket scheme | - | Sørensen et al. (2011); Simonsen et al. (2013) | Immediate runoff on top of an ice layer | Schwander et al. (1997) |
| GEUS | 200 layers dynamically allocated, Lagrangian | Darcy's law | Calonne et al. (2012), van Genuchten (1980) with coefficient from Hirashima et al. (2010) | Vionnet et al. (2012) | Darcy flow to adjacent cell given surface slope | Calonne et al. (2011) |
| DMIHH | 32 layers, Eulerian | | | | Zuo and Oerlemans (1996) | Yen (1981) |
| IMAU-FDM | maximum of 3000 layers, Lagrangian | Bucket scheme | - | Kuipers Munneke et al. (2015) | Only at the bottom of the column | Anderson (1976) |
| MeyerHewitt | finite volume, Eulerian, 600 layers | Darcy's law | Carman-Kozeny (Bear, 1972); Gray (1996) | Herron and Langway (1980) | Excess surface water | Meyer and Hewitt (2017) |
| UppsalaUniBucket | 600 layers, max 0.1 m layer thickness. Lagrangian | Bucket scheme | - | Ligtenberg et al. (2011) | Only at the bottom of the column | Sturm et al. (1997) |
| UppsalaUniDeepPerc | 600 layers, max 0.1 m layer thickness. Lagrangian | Deep percolation scheme; linear distribution down to 6 m (Marchenko et al., 2017) | | | | |

### 2.7. UppsalaUniBucket and UppsalaUniDeepPerc models

UppsalaUniBucket and UppsalaUniDeepPerc have been developed for the Norwegian Arctic (Van Pelt et al., 2012; 2019; Marchenko et al., 2017) and only differ in their representation of vertical water transport. UppsalaUniBucket simulates melt water percolation according to a bucket scheme while UppsalaUniDeepPerc uses a deep percolation scheme which mimics the effect of fast vertical transport due to preferential flow (Marchenko et al., 2017). This deep percolation scheme acts before the bucket scheme and instantaneously transfers the meltwater available at the surface to underlying layers using a linear distribution function of depth that reaches zero at 6 m depth (Marchenko et al., 2017). The water transport model incorporates irreducible water storage and allow infiltration through ice-dominated layers. All water that reaches the base of the firn column is set to run off instantaneously. References for the parameterizations used for gravitational settling, thermal conductivity, irreducible water storage and water percolation are given in Table 2.

## 3. Methods

### 3.1. Site selection and surface forcing

Differences between firn-model outputs and observations depend on the model formulation but also on the forcing data that are given to the model: any bias in forcing data propagate into the model output. To make sure we compare and evaluate the models independently of biases that may exist in forcing datasets that come from regional climate models, we use meteorological fields derived from five AWS at four sites.

These sites represent a broad range of climatic conditions on the Greenland ice sheet (Table 3, Figure 1) that produce a wide variety of firn density and temperature profiles. For example, the cold and dry climate at Summit Station produces cold firn with low compaction rates representative of the "dry snow" area as defined by Benson (1962). Dye-2, located in an area with higher melt (Table 3), is representative of the "percolation area" (Benson, 1962) where meltwater generated at the surface percolates into the firn and releases latent heat when refreezing into ice lenses. At the KAN_U site, lower accumulation rates and increasing melt have led to the formation of thick ice slabs (Machguth et al., 2016; MacFerrin et al., 2019) that impede meltwater percolation below 5 m. The Firn Aquifer (FA) site in Southeast Greenland has both high surface melt and high accumulation rate, leading to the formation of a perennial body of liquid water at a depth of 12 m and below (Forster et al., 2012; Kuipers Munneke et al., 2014).

We use data from the Greenland Climate Network (GC-Net) AWS at Dye-2 and Summit (Steffen et al., 1996) and from the Programme for Monitoring of the Greenland Ice Sheet (PROMICE) station at KAN_U (Ahlstrøm, et al., 2008; Charalampidis, et al., 2015) . For Dye-2 in 2016, we use a AWS installed by the University of Calgary and described in Samimi et al. (2020). Since this station was more recently installed than the GC-Net station, it ensures better meteorological

observations (levelling, absence of frost/mist on radiometers) and therefor better forcing for the models over the 2016 melting season, during which an extensive observational dataset is available for model evaluation. This simulation is henceforth

referred as Dye-2_16 while the longer simulation using the GC-Net AWS is referred as Dye-2_long. At the Firn Aquifer site, we use data from the S21 AWS maintained by Utrecht University. The S21 AWS measures air temperature, relative humidity (Vaisala HMP35AC), air pressure (Vaisala PTB101B), wind speed and direction (Young 05103), the shortwave and longwave radiative fluxes (Kipp and Zonen CNR1) as well as station tilt and instruments height. All quantities are sampled every 6 min, and hourly averages are recorded by a Campbell CR10X datalogger.

Data from each AWS were quality checked and obvious sensor malfunctions were discarded. No data were discarded at FA and Dye-2_16. Gaps in the temperature, wind speed, humidity, air pressure, incoming shortwave and longwave radiation, were filled with adjusted values from either nearby stations or HIRHAM5, following Vandecrux et al. (2018). Gaps in upward shortwave radiation were filled using gap-filled downward shortwave radiation and the nearest daily albedo values from the Moderate Resolution Imaging Spectroradiometer (MODIS) satellite (Box et al., 2017). Downward longwave radiation is not

monitored by the GC-Net stations (Dye-2_long and Summit) and is taken entirely from HIRHAM5 output.

The gap-filled meteorological fields are used to calculate the surface energy balance, based on the model developed by van As et al. (2005) and applied in Vandecrux et al. (2018). We use surface height measurements and available snow pit information to calculate snowfall rates as in Vandecrux et al. (2018). This surface energy and mass balance provides, at three-hourly resolution, the three surface forcing fields that were used by all models: the surface skin temperature, the amount of

meltwater generated at the surface, and net snow accumulation (snowfall – sublimation + deposition). Only the MeyerHewitt model required minor adaptation of these forcing fields (see Supplementary Text S1). Rain is not monitored at any site, so is not included in the mass fluxes. Tilt of the radiation sensor was not corrected for at Dye-2_long and Summit stations although this correction was seen to increase the calculated melt by 35 mm w.e. yr$^{-1}$ at Dye-2 (Vandecrux et al., 2020a). The surface forcing data are illustrated in Figure S3.

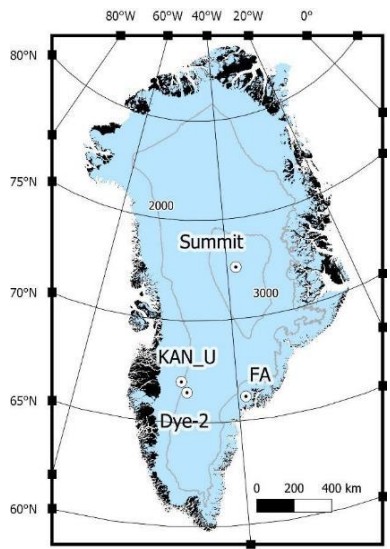


**Figure 1: Map of the four study sites. Elevation contours are in meters above sea level.**

**Table 3: Information about the four sites and five model runs considered in the comparison including mean annual accumulation ($\bar{b}$), mean annual air temperature ($\bar{T}_a$) and prescribed bottom firn temperature ($T_{bot}$).**

| Station name | KAN_U | Dye-2_long | Dye-2_16 | Summit | Firn Aquifer (FA) |
|---|---|---|---|---|---|
| Latitude (ºN) | 67.00 | 66.48 | 66.48 | 72.58 | 66.37 |
| Longitude (ºW) | 47.03 | 46.28 | 46.28 | 38.50 | 39.32 |
| Elevation (m a.s.l.) | 1840 | 2165 | 2165 | 3254 | 1663 |
| Surface slope (º) | 0.5 | 0.2 | 0.2 | 0 | 0.6 |
| Start date | 01 May 2012 | 01 June 1998 | 02 May 2016 | 02 July 2000 | 12 Apr. 2014 |
| End date | 31 Dec. 2016 | 02 May 2015 | 28 Oct. 2016 | 08 Mar. 2015 | 02 Dec. 2014 |
| $\bar{b}$ (mm w.e.) | 543 | 476 | 476 | 159 | 1739 |
| $\bar{T}_a$ (ºC) | -12 | -16 | -16 | -26 | -7 |
| $T_{bot}$ (ºC) | -9 | -15.5 | -13 | -31 | 0 |
| Initial firn density | Top 10 m: core_1_2012 (Machguth et al., 2016) From 10 to 60 m: Site J, 1989 (Kameda et al., 1995) | Dye-2 1998 core B (Mosley-Thompson et al., 2001) | Top 18 m: Core_10_2016 (Vandecrux et al., 2019) From 10 to 60 m: Dye-2 1998 core B (Mosley-Thompson, et al., 2001) | Top 8m: core from 1990 by Mayewski and Whitlow (2016) From 8 to 60 m: GRIP core (Spencer et al., 2001) | Top 8 m: FA-14 (Montgomery et al., 2018) From 8 to 60 m: FA-13 (Koenig et al., 2014) |

### 3.2. Boundary conditions

To allow fair comparison of the various firn models, as many boundary conditions as possible were specified in common for all models. A key parameter in firn models is the density of fresh snow added at the top of the model column. Here, all models used the value of 315 kg m$^{-3}$ from Fausto et al. (2018) which is derived from a compilation of 200 top 10 cm snow density observations from the Greenland ice sheet. Initial profiles for density, temperature and liquid water content (only at FA) were provided to all models and illustrated in Supplementary Figure S4. The references for the initial density profiles are given in Table 3. Initial temperature profiles were calculated using the first reading of air temperature (as first guess of

surface temperature), the first valid measurement of firn temperature, and the bottom firn temperature (Table 3). The bottom firn temperatures (Table 3), needed as lower boundary condition by some of the models, were calculated from the available firn temperature measurements. At KAN_U, the average of the deepest firn temperature, at ~8 m depth, was taken over spring 2013 – spring 2015 period. At Summit and Dye-2_long, the 10 m firn temperature was interpolated when firn temperature measurements were below 10 m depth and then averaged. For Dye-2_16 and FA, the deepest firn temperature measurement,


at 9 and 25 m depth respectively, were averaged over their respective measurement periods (Table 3). Initial liquid water content at FA is calculated according to the observations from Koenig et al. (2014) which indicate pore saturation below 12.2 m depth. Some models also need long-term mean air temperature and accumulation (Table 3) which were calculated from Box (2013) and Box et al. (2013).

**3.3.  Intercomparison and evaluation of model output**

Participating models provided simulated firn density, temperature and liquid water content at three-hourly time steps, interpolated to a common 10 cm grid from the surface to 20 m depth. Additionally, three-hourly vertically integrated refreezing and runoff were calculated by each model.

Three types of datasets are available at our sites for model evaluation: i) firn temperature observations from AWS as presented by Vandecrux et al. (2020a) at Summit and Dye-2_long, Heilig et al. (2018) at Dye-2_16 , Charalampidis et al. (2015) at KAN_U and Koenig et al. (2014) at the FA station; ii) firn density profiles (Table 4); and iii) observations of meltwater infiltration depth at Dye-2 from an upward-looking Ground Penetrating Radar (upGPR) during the summer 2016 (Heilig et al., 2018).

**Table 4. Firn cores used for model evaluation**

|        | Date          | Reference                   |
|--------|---------------|-----------------------------|
|        | 5 March 2001  | Dibb and Fahnestock (2004)  |
| Summit | 1 July 2007   | Lomonaco et al. (2011)      |
|        | 29 May 2015   | Vandecrux et al. (2018)     |
|        | 17 April 2011 | Forster et al. (2014)       |
| Dye-2  | 5 May 2013    | Machguth et al. (2016)      |
|        | 21 May 2015   | Vandecrux et al. (2018)     |
|        | 1 May 2012    | Machguth et al. (2016)      |
| KAN_U  | 27 April 2013 |                             |
|        | 5 May 2015    | MacFerrin et al. (2019)     |
|        | 28 April 2016 |                             |

For firn density, we calculate for each time step the average firn density over the 0-1 m, 1-10 m and 10-20 m depth ranges and discuss the standard deviation of these values among models and their deviation from firn core observations. We also compare the simulated density profiles to the firn core data at each site. For firn temperature, we compare hourly observations

of firn temperature to interpolated temperature from the closest model layers and use the Mean Error (ME), Root Mean Squared Error (RMSE) and coefficient of determination ($R^2$) to quantify the performance of the models with respect to the observations.

## 4. Results

In the following, we present comparisons of firn model outputs and model deviations from observations for firn temperature, density, and liquid water content at sites representing different firn and meltwater regimes: dry firn (Summit), the percolation zone (Dye-2), ice slabs (KAN_U), and a firn aquifer (FA site).

### 4.1. Dry firn site: Summit

At Summit, density evolves in a similar manner across all models: low density snow is deposited at the surface and is advected to greater depth (Figure 2a). All models except DMIHH and MeyerHewitt preserve the layering of the initial density profiles as it gets advected downward and generate layered firn at the surface. The temporal evolution of the average density for the 0-1 m depth range follows similar seasonality and slight increasing trend (Figure 3a). Over the 1-10 m and 10-20 m depth (Figure 3b,c), most models produce increasing firn density apart from IMAU-FDM in which the firn density slightly decreases. All models agree relatively well on the average density independent of the depth range, with a maximum standard deviation among models of 15 kg m$^{-3}$ for the top 1 m average density (of 336 kg m$^{-3}$), 27 kg m$^{-3}$ for the 1-10 m range (420 kg m$^{-3}$ on average) and 23 kg m$^{-3}$ for the 10-20 m range (542 kg m$^{-3}$ on average) during the 15-years-long simulation period (Figure 3). In comparison with the firn cores drilled in 2007 and 2015, most models reproduce vertical variability in firn density within observation uncertainties (Figure 3d,e). The evaluation of the density profile reveals that IMAU-FDM underestimates firn density between 5 and 15 m depth.

Regarding firn temperature, in most models, seasonal skin temperature fluctuations drive firn temperature variability in the top few meters of the column. However, seasonal temperature fluctuations propagate much deeper in the DTU model while it is almost not visible in MeyerHewitt model (Figure 2b). This results in much lower R$^2$ when comparing these two models to firn temperature observation: 0.41 and 0.28 for DTU and MeyerHewitt respectively. This results from the numerical strategy and/or thermal diffusivity used in these models. Models that have explicit formulation for deep meltwater infiltration (CFM-Cr, CFM-KM and UppsalaUniDeepPerc) have positive ME 0.6 to 0.7 °C. This is due to the simulation of short-lived deep percolation events that infiltrates the minor melt from the surface down to ~5 m, and to the subsequent refreezing and latent heat release. DMIHH, GEUS, IMAU-FDM and UppsalaUniBucket provide the lowest ME compared to firn temperature observations (Figure 2c). Yet, it should be noted that IMAU-FDM calculates adequate heat diffusion while underestimating the firn density (Figure 3e). Either the firn density underestimation in IMAU-FDM is not sufficient to induce a noticeable change in thermal conductivity or the thermal conductivity and/or numerical scheme used by IMAU-FDM compensate for the underestimated density and result in adequate simulated firn temperature.

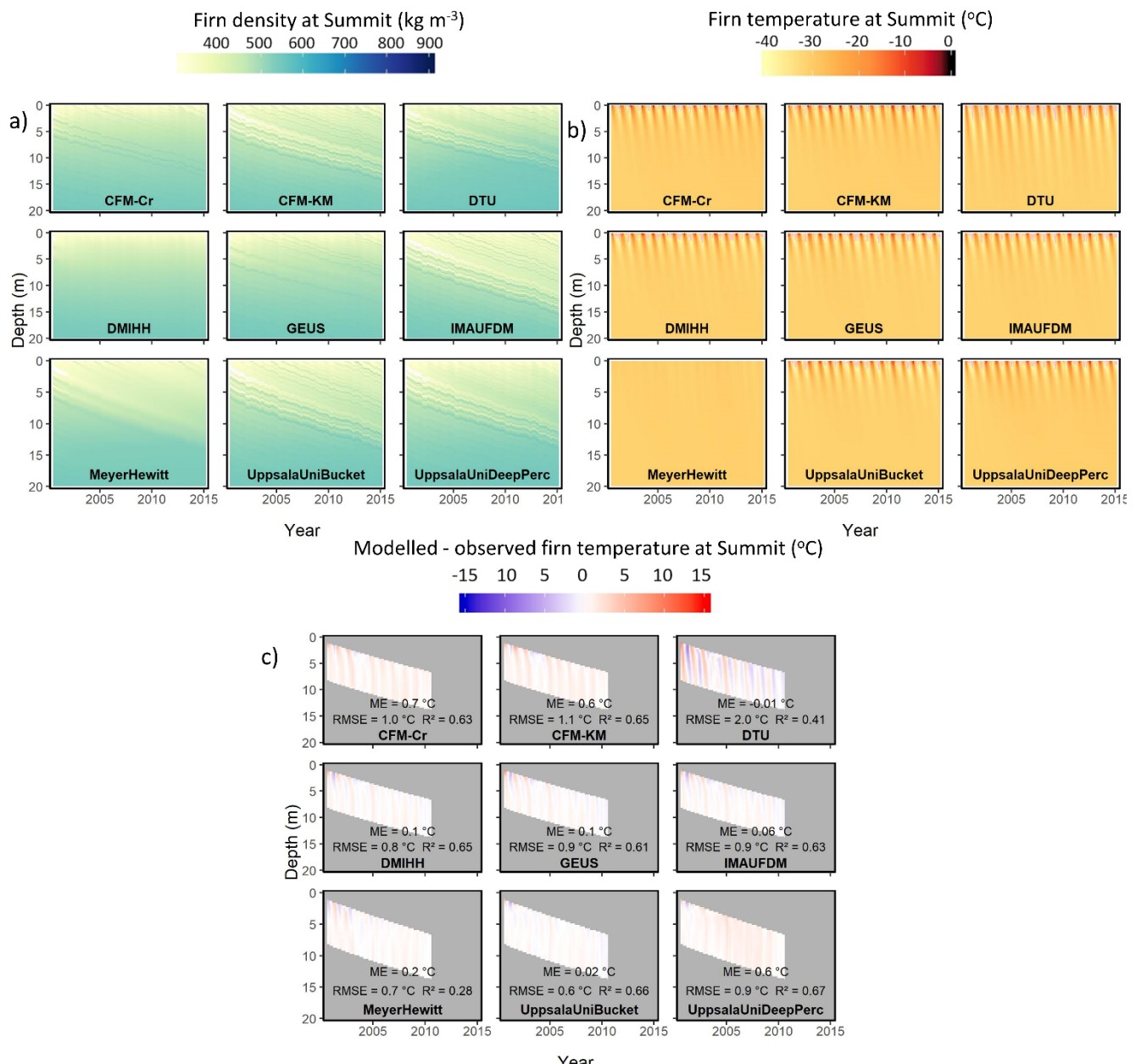

**Figure 2: Simulated firn density (a), temperature (b) and deviation between simulated and observed firn temperature (c) at Summit.**

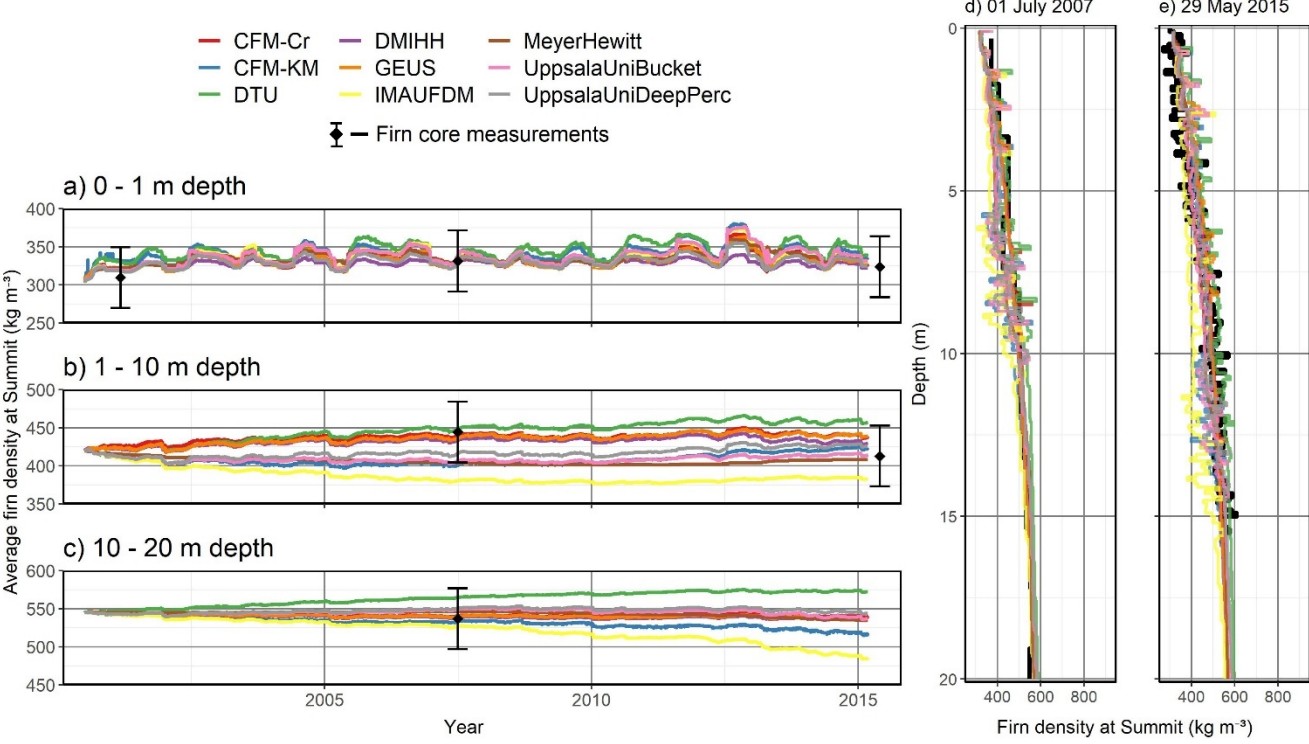

**Figure 3: Modelled (colored lines) and observed (black dots with ±40 kg m-3 uncertainty bars) average firn density for the 0-1 m (a), 1-10 m (b) and 10-20 m depth range (c) at Summit. Note the different density scales. Comparison of simulated and observed firn density profiles (d, e). In (e) the last modelled density profile, from 8 March 2015, is compared to an observation from 29 May 2015.**

### 4.2. Percolation site: Dye-2

At Dye-2 surface melt occurs every summer. Consequently, refreezing of percolating meltwater has a significant effect on simulated density and temperature (Figure 4). The investigated models span a large spectrum of meltwater infiltration strategies (Table 2), leading to greater differences between models in firn density, temperature and liquid water content (Figure 4). Simulated meltwater percolation depth varies greatly among the models (Figure 4c). At one end of the spectrum, the DTU model only allows meltwater in the top model layer; an ice layer is built right at the start of the simulation and water is not able to penetrate ice layers in this model. At the other end, CFM-Cr and CFM-KM, which do allow meltwater to pass through ice layers and explicitly account for fast preferential flow, simulate percolation down to 10 m depth. In between these end-member models, UppsalaUniDeepPerc simulates percolation, up to ~5 m depth. IMAUFDM, UppsalaUniBucket, DMIHH and GEUS models give similar results and percolate water down to 1-3 m.

These differences in meltwater infiltration, when accumulated over a 17-years-long run, lead to large differences in firn density and temperature evolution across models (Figure 4). Models that include deep water infiltration (CFM-Cr, CFM-KM

and UppsalaUniDeepPerc) build up a thick high-density layer at 3-10 m depth. In contrast, DTU, GEUS, IMAUFDM and UppsalaUniBucket simulate thinner, high-density, layers that form each summer at the surface and are buried in the following months and years. These sharp contrasts between low- and high-density layers are smoothed in the Eulerian DMIHH and MeyerHewitt models. For each model, the simulated firn temperature at Dye-2 (Figure 4b) and its deviation from observations (Figure 4d) responds closely to the simulated meltwater infiltration each summer (Figure 4c). Models that include explicitly deep percolation (CFM-Cr, CFM-Kr, UppsalaUniDeep) also present the greatest firn warming at depth, due to refreezing and latent heat release (Figure 4b), and consequently have a positive ME ranging from 3.6 °C to 6.2 °C (Figure 4d). The DTU model does not percolate meltwater deep into the firn (Figure 4c) and consequently firn temperature evolves only due to heat diffusion, which leads to a cold bias (ME = -1.6 °C, Figure 4d). The remaining models (DMIHH, GEUS, IMAU-FDM, UppsalaUniBucket and MeyerHewitt) simulate similar inter-annual variability in meltwater infiltration and similar performance in firn temperature with a ME within ± 1 °C and $R^2 > 0.5$.

The impact of these different infiltration patterns on the long-term evolution of the average firn density and how simulated firn density compares to observations are presented in Figure 5. The standard deviation (model spread) of density reaches 161 kg m$^{-3}$ in the top meter of firn and 141 kg m$^{-3}$ for the 1-10 m layer (Figure 5). Lower deviation (29 kg m$^{-3}$) between 10-20 m stems from the limited time span of the simulation that does not allow the advection of the portion of firn where models disagree below 10 m depth (Figure 4 and 5).

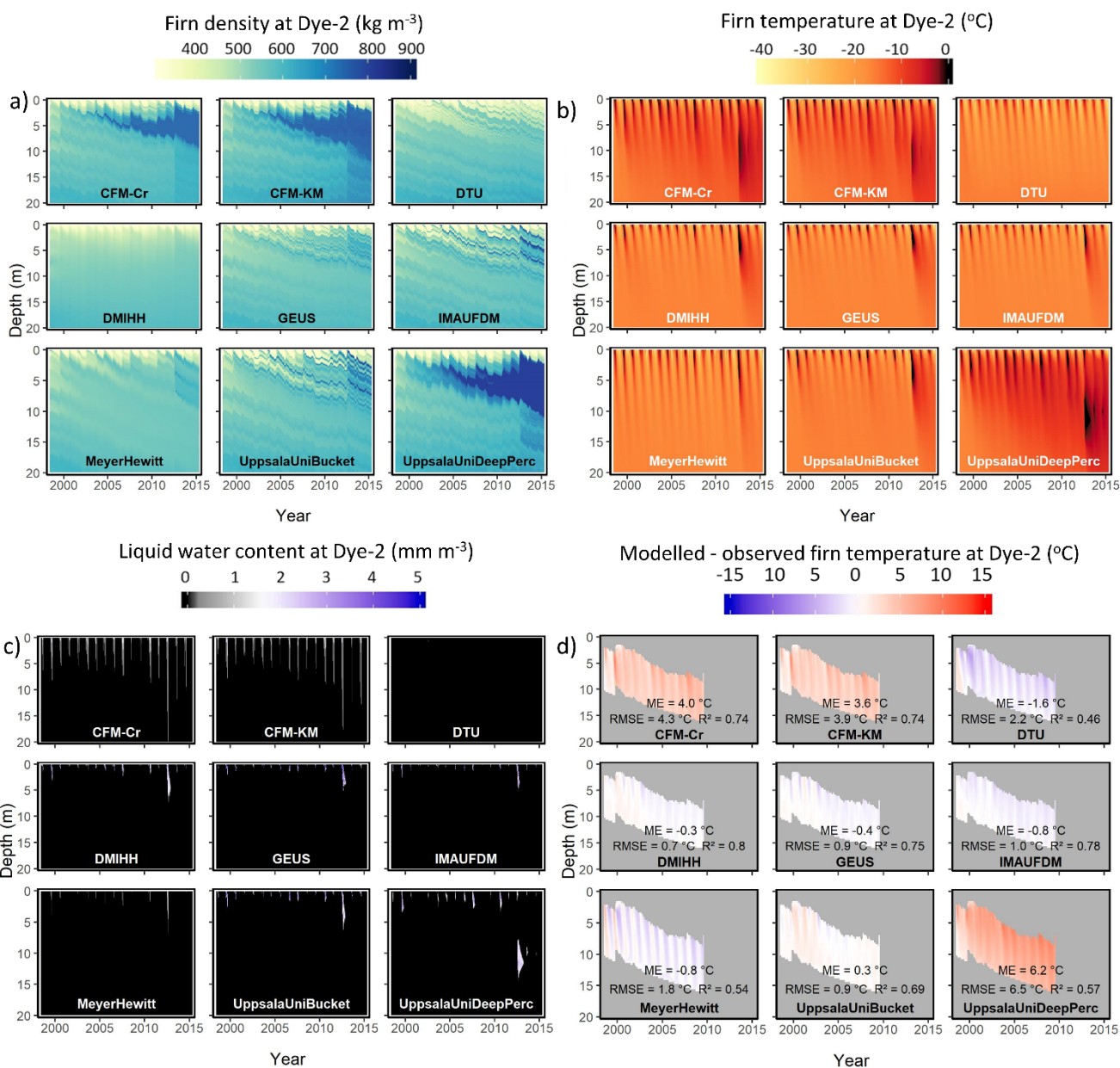

Figure 4: Simulated firn density (a), temperature (b), water content (c) and deviation between simulated and observed firn temperature (d) at Dye-2_long.

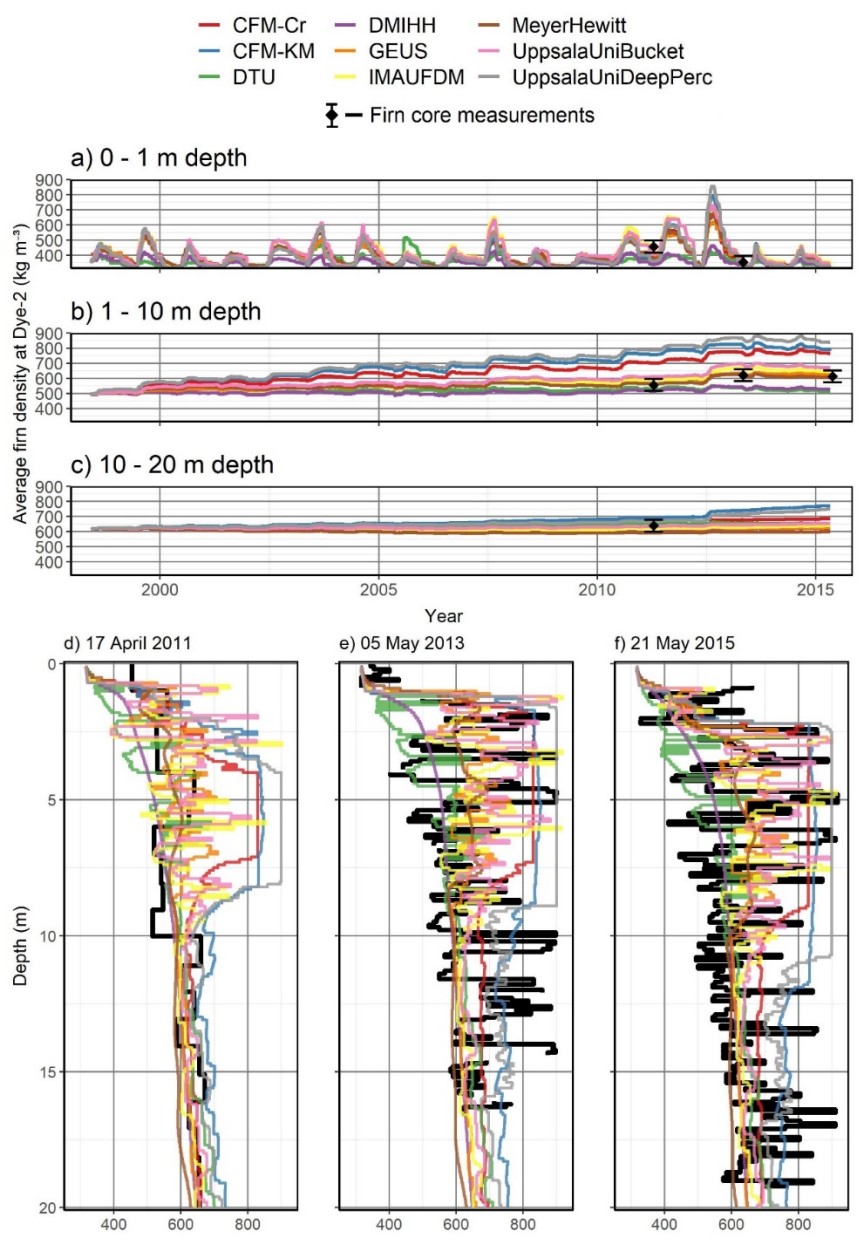

**Figure 5: Modelled (colored lines) and observed (black dots with 40 kg m-3 uncertainty bars) average firn density for the top 1 m (a), for the 1-10 m depth range (b) and 10-20 depth range (c) at Dye-2_long. Observed and simulated vertical variability in density at Dye-2_long (d-f).**

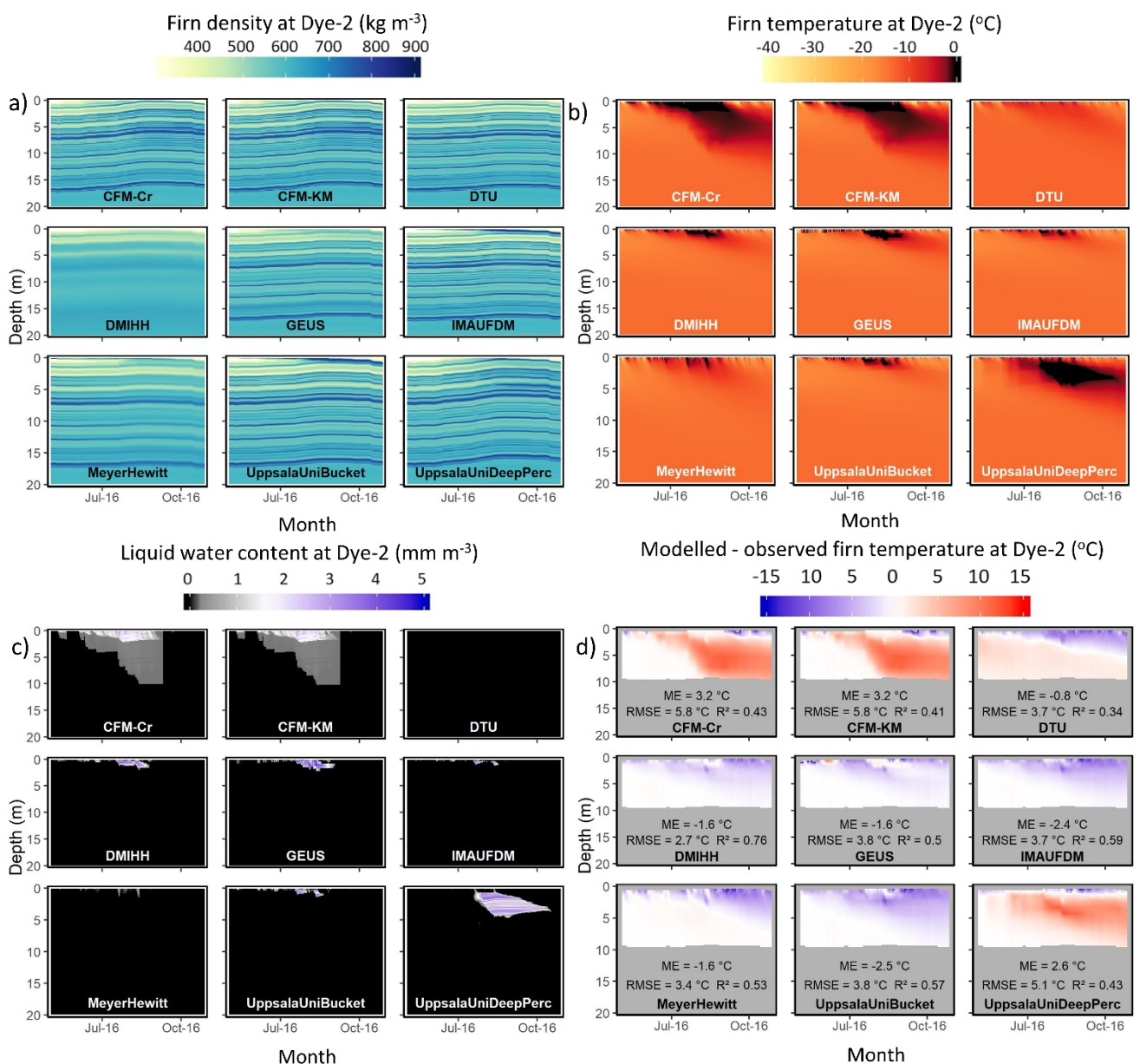

**Figure 6: Simulated firn density (a), temperature (b), water content (note different y-axis) (c) and deviation between simulated and observed firn temperature (d) in Dye-2_16.**

The use of a more recent AWS to derive the climate forcing at Dye-2_16 allows the assessment of the firn models and their infiltration schemes in the best conditions. Over a single melt season, the meltwater infiltration and refreezing does not produce

355   drastic changes in the simulated density profiles (Figure 6a). Yet, the meltwater is distributed at different depths and with different timing depending on the model (Figure 6c). The dual-domain approach of CFM-Cr and CFM-KM is visible with higher liquid water content close to the surface, corresponding to the matrix flow, and low water content infiltrating down to 10 m depth in the heterogenous percolation domain. UppsalaUniDeep, which also includes deep percolation, infiltrates water down to ~5 m, deeper than the models using a parametrization of Darcy's law (DMIHH, GEUS models) and bucket scheme

360   (IMAU-FDM, UppsalaUniBucket models) which do not show liquid water below ~2 m depth (Figure 6c). As a result of these differences in meltwater infiltration and location of the meltwater refreezing, the firn temperature differs from model to model (Figure 6b). The deep percolation models (CFM-Cr, CFM-KM and UppsalaUniDeep) have a marked positive bias (ME>2.6 °C). The DTU model, which does not infiltrate water below the first few layers show a cold bias in the top part 5 m of the firn, where all the other models simulated meltwater infiltration. All the other models simulate colder conditions than observed

365   with ME ranging from -2.5 °C in UppsalaUniBucket to -1.6 °C in the GEUS model.

UpGPR observations (Figure 7) show that the meltwater did not reach below 2.5 m depth during the 2016 melt season. The melt was concentrated around three periods of increasing intensity between May and June and a period when meltwater was continuously present in the firn between 20 July and 25 September. Compared to the upGPR, the CFM-CR and CFM-KM

370   models substantially overestimate percolation depth (Figure 7a, red and blue lines), suggesting that, in the current configuration, these models exaggerate the effects of preferential flow, at least at this location. The DTU model does not simulate any percolation, and the MeyerHewitt model simulates the presence of meltwater in short-lived, episodic pulses rather than the continuous presence of meltwater that the upGPR observed. The other models simulate a percolation depth and temporal behavior closer to the upGPR observations.

375

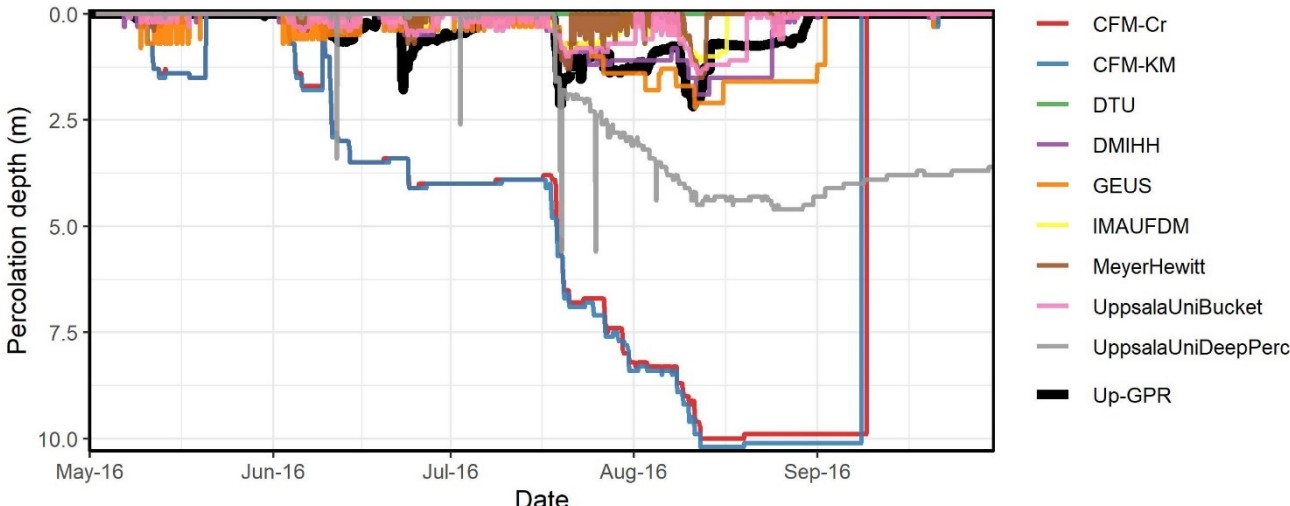

**Figure 7: Comparison of the simulated (colored lines) and upGPR-derived (black line) meltwater percolation depth at Dye-2 over the 2016 melting season.**

### 4.3. Ice-slab formation: KAN_U

At KAN-U, surface melt is more intense than at Dye-2. As a result, refreezing of infiltrated meltwater forms ice slabs that can be tens of centimeters to several meters thick. This site is therefore an interesting test for the firn models to see how they handle the presence of an ice-slab, and the effects of ice slabs on the vertical profiles of temperature and liquid water. Note that the firn models are initialized in spring 2012 with a pre-existing ice slab, which means that we do not assess the model capacity to form an ice slab: we only assess the effect of the ice slab on the evolution of the firn column.

The evolution of the density profile at KAN_U strongly depends on whether the model allows percolation past the ice slab (Figure 8a,c). The DMIHH, MeyerHewitt and DTU models do not allow such percolation at all, and thus refreezing-related densification only occurs on top of the ice slab. The absence of latent heat release below the ice slab causes these models to exhibit colder temperatures than observed (Figure 8b,c). Another group of models (CFM-Cr, CFM-KM, IMAUFDM, UppsalaUniBucket and UppsalaUniDeepPerc) does allow for percolation of meltwater through the ice slab, to depths of 10-15 m. As a result, the small amount of available pore space within the ice slab is used for refreezing and progressively filled (Figure 8a). Nevertheless, the sealing of the ice slab in these models does not prevent the meltwater from percolating through, and meltwater refreezing continues to occur at depth and to densify the firn there. These models overestimate deep firn temperatures compared to observations (Figure 8d), presumably as a result of excess refreezing. In the MeyerHewitt and DMIHH models, the initial ice layers are gradually smoothed over time (Figure 9d-f). We relate this behavior to their Eulerian framework that implies frequent averaging of firn density and temperature when mass is added or removed from the model column. Still, they keep higher density between 5 and 10 m depth where the ice slab is. The model spread in top 1 m average

density is minimal in the spring and increases in the summer (Figure 9a). The simulated average densities for 0-1, 1-10 and 10-20 m depth ranges compare well with punctual observations (Figure 9a-c) but deviations between simulated and observed density profiles increase with time (Figure 9d-f). Comparison of the simulated firn temperature to hourly observations confirms that models including deep percolation (CFM-Cr, CFM-KM and UppsalaUniDeep) and bucket schemes (IMAUFDM and UppsalaUniBucket) infiltrate too much water at depth, resulting in a positive bias in temperature and a ME ranging from 1.8 to 4.7 °C. The DTU and MeyerHewitt models do not show any meltwater infiltration or latent heat release at depth (Figure 8b,c). Consequently, they show lower firn temperature than observed with ME of -5.3 and -3.6 °C respectively. The GEUS model uses a low, but not null, permeability to ice layers and thus simulates reduced infiltration through the ice slab (Figure 8c) which leads, after this water refreezes, to firn temperatures closest to observations (ME = 0.6 °C).

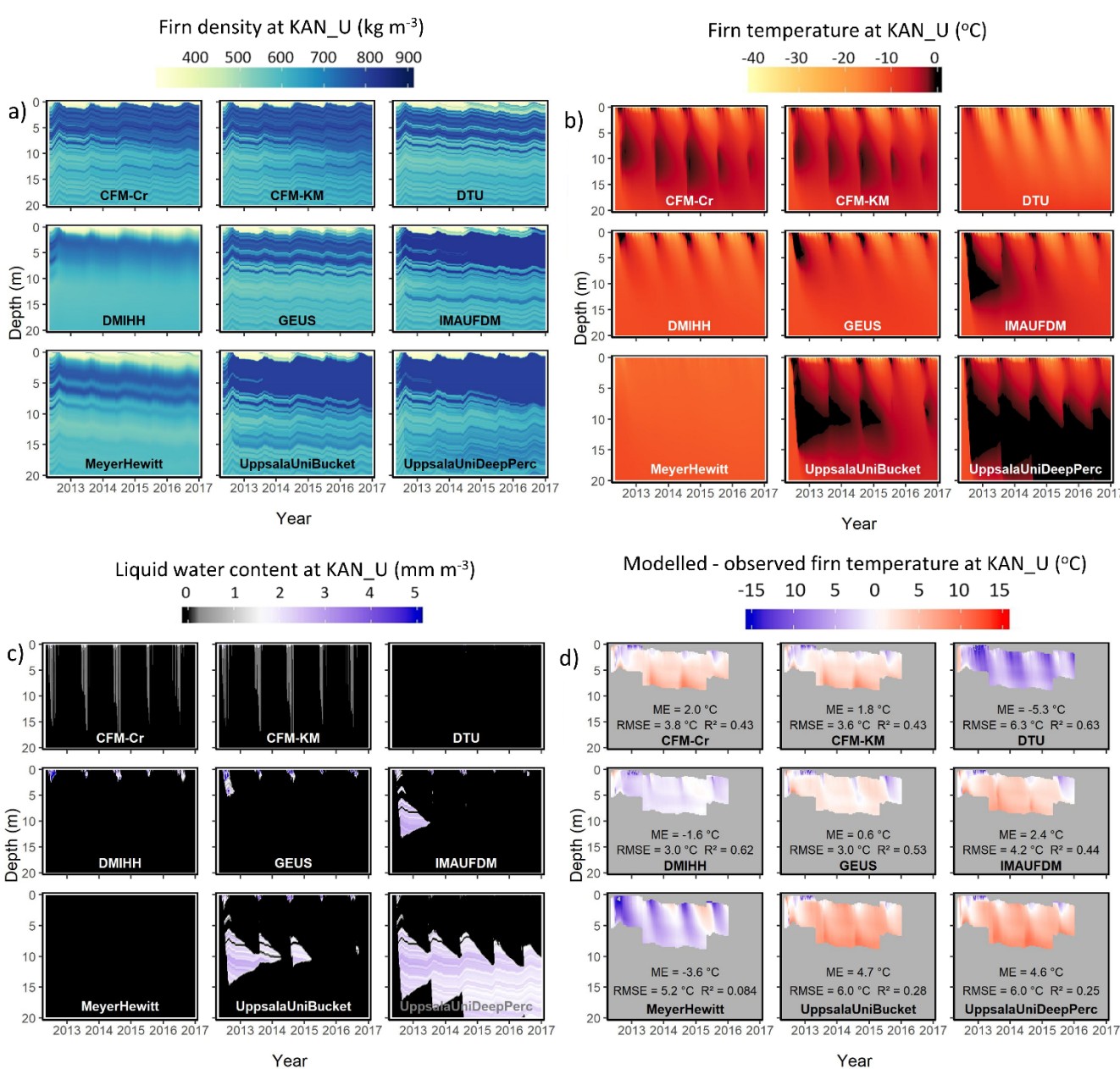

410

**Figure 8: Simulated firn density (a), temperature (b), water content (note different y-axis) (c) and deviation between simulated and observed firn temperature (d) at KAN_U.**

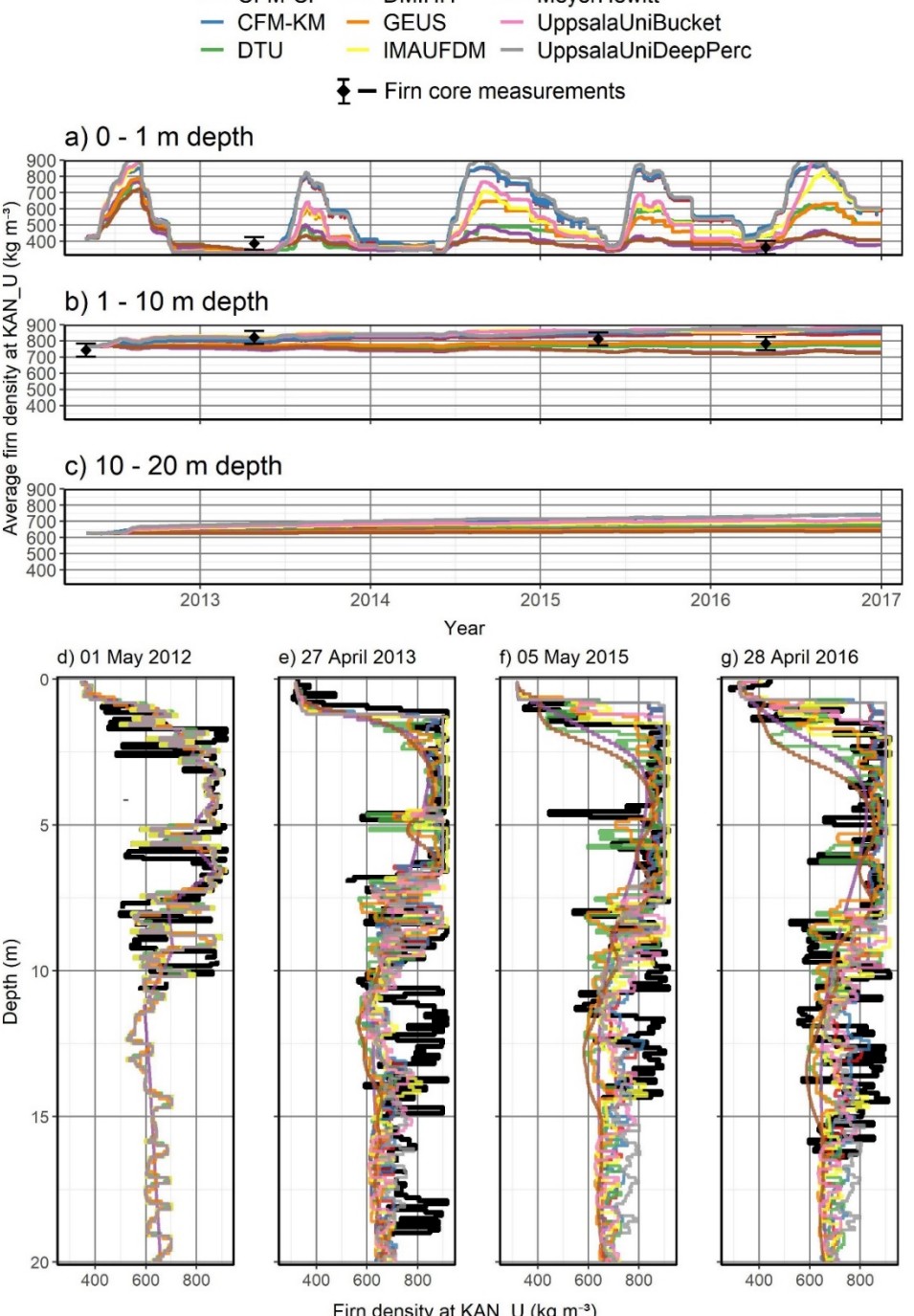

Figure 9: Modelled (colored lines) and observed (black dots with 40 kg m-3 uncertainty bars) average firn density for the top 1 m (a), for the 1-10 m depth range (b) and 10-20 depth range (c) at KAN_U. Observed and simulated density at KAN_U (d-f).

### 4.4. Firn aquifers: FA site

At the firn aquifer site, both melting and snowfall are high, leading to perennial storage of liquid water within the firn pack. In terms of firn density, vertical gradients are similar among models but both the MeyerHewitt and DMIHH models simulate smoother profiles (Figure 10a). This is likely due to their use of an Eulerian framework, as also seen in the results for KAN-U. Temporal evolution in density is also similar among models given the short span of the simulation. The DTU model simulates slightly denser firn in the top few meters of the column as a result of refreezing (Figure 10a). Models which account for preferential flow (both CFM models and UppsalaUniDeep) simulate meltwater infiltration to the aquifer, although with a slight difference in timing (Figure 10b,c). Unfortunately, the firn temperature observations do not allow us to ascertain how much water was transferred to the aquifer, but only that the whole firn column was at 0 °C from mid-August to late September 2014, when cold surface temperature started to diffuse into the firn. These three deep percolation models overestimate shallow firn temperature in summer, and underestimate shallow firn temperature in winter, when compared to observations (Figure 10d). In the absence of ice layers within the upper firn, the DTU model simulates fast meltwater infiltration through the top 12 m and thus simulates a firn column entirely at 0 °C (Figure 10b), in accordance with firn temperature observations (Figure 10d), but this meltwater runs off shortly after it percolates (Figure 10c). The other models simulate a firn column that is slightly too cold with ME between -0.1 and -0.6 °C. As a result of the prescribed liquid water at depth in the initial conditions, deep firn temperature remains at melting point year-round in all models (Figure 10b), with liquid water at depth in all models except DTU (Figure 10c).

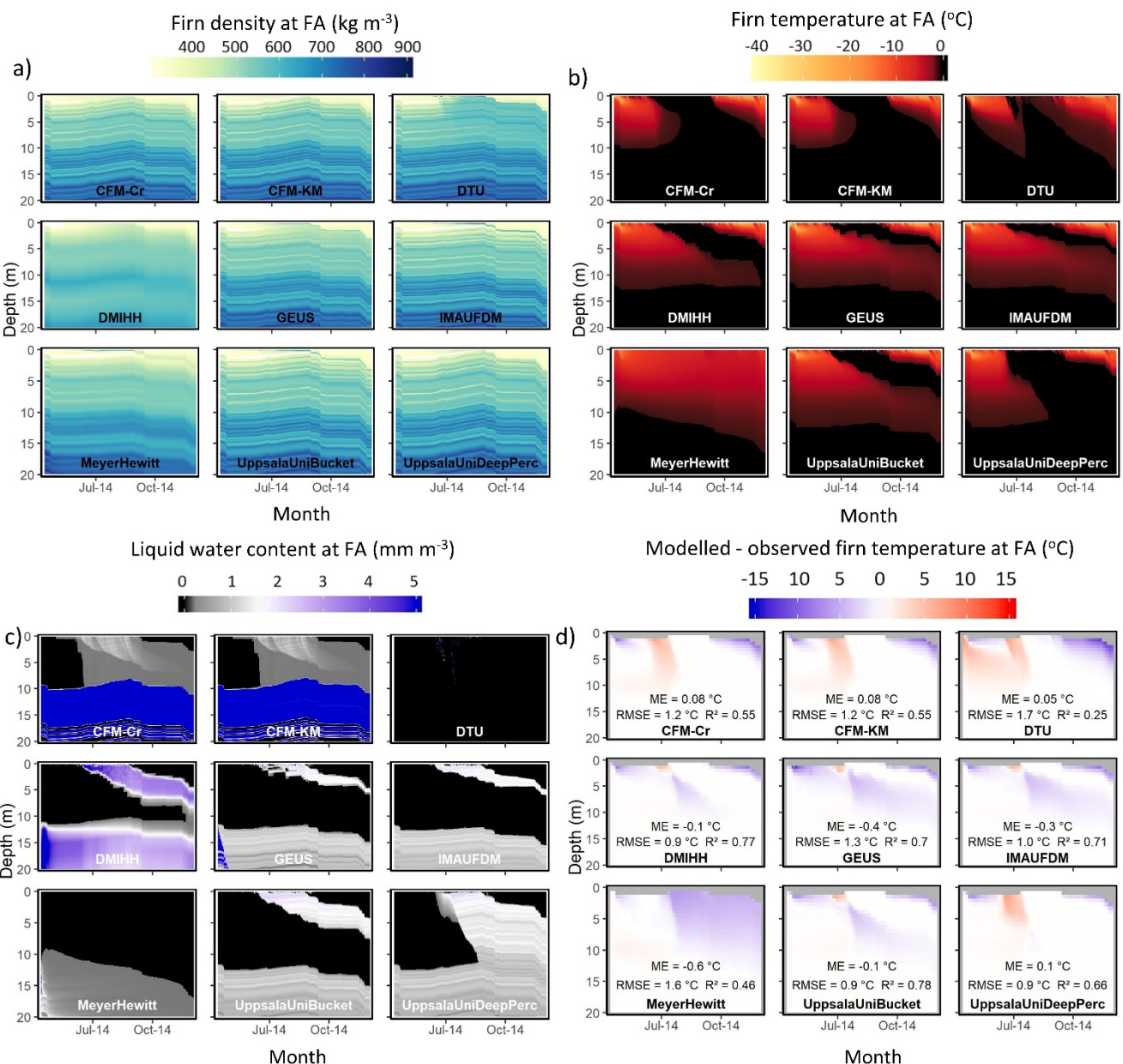

**Figure 10: Simulated firn density (a), temperature (b), water content (c) and deviation between simulated and observed firn temperature (d) at FA.**

## 5. Discussion

The variability in simulated firn density, temperature, and water content among the models, and the deviation between simulations and observations (Section 4) can be explained by the various ways physical processes are accounted for in the models. In this section we detail what can be learned from the comparison and we explore current knowledge gaps and potential improvements for firn models.

### 5.1. Dry firn and heat transfer

At Summit, comparisons with observations suggest that with appropriate forcing, the various densification formulations perform similarly and within observational uncertainty. The ability of firn models in the dry snow area to reproduce measured density profiles has been established from previous comparisons (Steger et al., 2017; Alexander et al., 2019), and can be attributed to the fact that most densification schemes are calibrated against firn density profiles from dry snow areas. The simulated densities at Summit show that densification schemes provide similar outputs, despite modelled temperatures being slightly different (Figure 2a,b). Still, the ability of firn densification models to simulate firn changes in a transient climate is less certain (Lundin et al., 2017), and should remain a priority for future study. We also note that densifications schemes developed for dry firn are applied to wet-firn zones, and further research is needed to determine the validity of this assumption.

At Summit, the top of the initial firn density profile is advected to 10 m depth by the end of the simulation (Figure 2). Consequently, we here assess both the models' capacity to accommodate and transform new snow at shallow depth and how models densify the initial density profile as it is advected downward. The persistence of the initial conditions consequently influences the performance of the models but have the advantage of giving all models the same starting point. An alternative strategy would have been to allow models to equilibrate with the surface forcing during a spin-up period. But such an approach would initiate models with their own spin-up result and would make more difficult to assign differences in model outputs to either model design or different initial conditions. Our observation-based initialization was therefore deemed more suitable to intercompare the meltwater retention in different models. In spite of measurement uncertainty and firn spatial heterogeneity, the firn density and temperature measurements used to initialize and evaluate the models represent the closest estimation of actual firn characteristics. Additionally, important biases in initial firn density and temperature would lead to a visible adjustment of the simulated firn characteristics in the first months/years as the model reacts to the surface forcing. No abrupt change can be seen in the simulations (Figure 2), which gives confidence that the initial conditions were appropriate.

Models exhibit small but clearly discernible differences in firn temperature at Summit (Figure 2b). In our model experiments, downward advection due to accumulation was identical for all models, suggesting that this spread must be caused by the parameterization of thermal conductivity and/or the models' differing numerical schemes. Also, a suite of models exhibits

colder temperatures compared with observations at Summit (DTU, DMIHH, GEUS, IMAUFDM, UppsalaUniBucket). We interpret this as an indication that heat transfer through the firn is still not accurately handled in most firn models. The heterogeneous nature of the firn, the presence of vertical ice features in the firn, the variability in surface snow density/thermal conductivity as well as firn ventilation are processes that are over-simplified or absent in the models and should be the subject of future research. Errors due to inaccurate estimates of thermal conductivity affect firn temperature, densification rates, and meltwater refreezing potential. We recommend that further work investigates potential improvements of the parameterization of thermal conductivity, either using recent studies (e.g., Calonne et al., 2019, Marchenko et al., 2019) or model calibration to observed firn temperature at dry firn locations. Other causes of model-data mismatch could be that certain processes (e.g. radiation penetration or variable fresh snow density) are not provided to the models, or that uncertainty in the forcing data derived from AWS observations will propagate into the model simulations.

## 5.2. Meltwater percolation and refreezing

Many observational studies have demonstrated that there are two pathways for meltwater to infiltrate into the firn, namely by homogeneous wetting front, also called matrix flow, and by preferential flow through vertically extended channels (e.g. Marsh and Woo, 1984; Pfeffer and Humphrey, 1996). Some of the nine participating firn models do include both percolation regimes, and others do not. The lack of preferential flow routines has recently been described as a limitation of firn models (e.g. van As et al., 2016). Yet, little is known about how often this phenomenon occurs in the firn, how deep meltwater is transported, and which process triggers preferential flow. Here, the models that explicitly include deep percolation (CFM-Cr, CFM-KM and UppsalaDeepPerc) overestimate percolation depth and firn temperature at Dye-2, KAN_U and even Summit where the surface meltwater production is minimal. In their current configurations, the deep percolation schemes seem less adapted for areas with minor melt. Our results suggest that until the physics of preferential flow in firn are better understood, these more-complex models do not necessarily provide better results than simple bucket schemes. We recommend targeted field campaigns and laboratory studies to better understand preferential flow, and using those to constrain under which firn conditions and meltwater input deep percolation occurs. These steps are necessary to develop accurate deep percolation schemes in firn models.

On the other hand, models that keep meltwater close to the surface, because they do not include any form of deep percolation, do not always show better performance. At Dye-2_16, DTU, DMIHH, GEUS, IMAUFDM and UppsalaUniBucket all exhibit temperatures that are too cold compared with the observations. The cold bias could be due partly to an underestimation of thermal conductivity (section 5.1), or to insufficient meltwater percolation. The upGPR observations at Dye-2 in 2016 indicates a reasonable percolation depth for all these models except DTU. It is conceivable that these models do simulate a reasonable percolation depth, but that the volume of percolating and refreezing meltwater is underestimated. Firn temperature observations and upGPR measurements can detect the presence of liquid water, but currently, no technique allows the vertically resolved measurement of water content. The models that use Darcy's law (CFM-Cr, CFM-KM, DMIHH, GEUS,

505 MeyerHewitt) use different formulations for the firn permeability (Table 2) which also contribute to differences in meltwater percolation and refreezing results. Firn permeability can be related to grain size and firn density (Calonne et al., 2012). However, firn grain size and permeability observations are scarce, and these variables remain totally unconstrained in current models. Future model evaluation should include the existing data where available (e.g. Albert and Shultz, 2002) and more field observations of these grain-scale characteristics should be collected.

## 510 **5.3. Ice slabs**

The formation of ice slabs is a complex interplay between accumulation, densification, meltwater percolation, and refreezing (Machguth et al., 2016). Simulation of ice slabs by a firn model is therefore highly challenging, and success or failure to reproduce ice slabs depends on a number of processes that are closely linked and difficult to disentangle. Models that include deep percolation (CFM-Cr, CFM-KM and UppsalaUniDeepPerc) grow an ice layer of several meters thickness close to the 515 surface at Dye-2, where no such ice slabs are observed. This model behavior can be explained by the simulation of water percolation bypassing ice layers and thus refreezing in cold underlying firn. At KAN_U, where ice slabs do exist, the DMIHH and GEUS models predict firn temperatures closest to the observations (lowest RMSE and highest $R^2$ for the DMIHH, lowest ME for GEUS) when compared to observations (Figure 8d). The performance of DMIHH at KAN_U can be explained by the absence of meltwater infiltration below the ice slab (Figure 8c) which agrees with recent field evidence of the ice slabs' 520 impermeability (MacFerrin et al., 2019). In DMIHH, the blocking of percolation originates from a simple permeability criterion: a layer reaching 810 kg m$^{-3}$ density becomes impermeable, and any incoming meltwater is sent to runoff. The choice of this value was based on work in Antarctica which found that firn permeability reaches zero over a range of densities centered on 810 kg m$^{-3}$ (Gregory et al., 2014). Unfortunately, such studies remain scarce in Greenland and results do not provide a definite constraint on permeability (e.g. Albert and Schulz, 2002; Sommers et al., 2017). The DTU model uses a 525 similar threshold density to characterize a layer's impermeability but found that 917 kg m$^{-3}$ gave the best match with observed firn density profiles (Simonsen et al., 2013). On the contrary, the IMAU-FDM model assumes that, at the horizontal resolution on which it usually operates (1-25 km$^2$), ice layers can be assumed to be discontinuous and are therefore never impermeable. We note that the ice slab has a low, but not null, permeability as illustrated by rarely observed meltwater refreezing events within the ice slab (Charalampidis et al., 2016). Unfortunately, few observations are available to evaluate 530 the effective permeability of ice slabs, both at local and regional scales and either confirm or contradict some of the assumptions made by the models. We recommend further investigation of the permeability of ice-dominated firn in relation to the firn density, the ice layer thickness and the various spatial and temporal scales at which the firn models are used.

Two models with a bucket-type percolation scheme, IMAUFDM and UppsalaUniBucket both use an irreducible water 535 content formulation established by Coléou and Lesaffre (1998) from laboratory measurements. They consequently present similar and realistic percolation depths at Dye-2 (Figure 4, 6 and 7). At KAN_U, however, in the presence of an ice slab, the two bucket-scheme models both overestimate percolation: this is evident from a warm bias there, relative to the firn

temperature observations (Figure 8). We therefore conclude that bucket schemes perform relatively well in the absence of ice slabs, and that they could benefit from an improved representation of flow-impeding ice layers.


Finally, we make a note on discretization strategies of firn models. In Lagrangian models, the numerical grid follows the firn as layers get buried under accumulating snow. In Eulerian models the firn is being transferred through a fixed numerical grid. The Eulerian models, DMIHH and MeyerHewitt, smooth the firn density profile, reducing and dissipating contrasts in firn density (Figures 2, 4 and 8). This smoothing is not prevented by increased vertical resolution since MeyerHewitt has 18

times more layers than DMIHH. At KAN_U, these two models gradually lose the contrast between the layers that compose the ice slab and the firn below (Figure 8). Therefore, Eulerian models tend to represent ice slabs in terms of a depth range with increased density, rather than marked layers of ice. This limitation of Eulerian models does not prevent the DMIHH model from simulating adequately firn temperature at KAN_U (Figure 8d) and water infiltration at Dye-2 (Figure 6). Further testing of Eularian models should investigate how this smoothing affects the modelled firn characteristics over longer runs

and how ice slabs are represented in these models.

## 5.4. Firn aquifers

Like ice slabs, firn aquifers form in locations with a complex combination of accumulation, surface melt, percolation, and refreezing (Forster et al., 2014; Kuipers Munneke et al., 2014). Both the thermodynamic and the hydrological components of a firn model play an important role in its capacity to simulate firn aquifers.


As a general observation, aquifers are poorly represented in the firn models considered in this intercomparison, which poses the question of the suitability of the models to simulate aquifers in Greenland. For example, horizontal water flow at depth plays a crucial role in the evolution of firn aquifers (Miller et al., 2018). However, the nine models investigated here, and to our knowledge all firn models currently used to evaluate surface mass balance on the Greenland ice sheet, are one-

dimensional. As such, the water available for lateral movement in these models is sent to runoff, which is itself governed by poorly constrained parameterizations. Also, IMAUFDM and UppsalaUniBucket do not allow for the presence of water beyond the irreducible water content: after the initialization of these models, all the excess water within the aquifer is run off instantaneously. As a result, these models are incapable of modelling actual aquifers (defined as saturated firn). Still, the regional climate model RACMO2, which includes IMAUFDM, has been used previously to map aquifers over the entire ice

sheet (Forster et al., 2015). Areas where the model showed residual subsurface water (within the irreducible water content) remaining in spring was assumed to represent areas where firn aquifers might be present. Although this approach succeeded at mapping the current firn aquifer areas, the difference between what is tracked in the model and what actually happens at firn aquifer puts doubt on the current capacity of firn models to predict firn aquifer evolution in future climate. Other models show an intermediate type of behavior: the DMIHH model runs off excess water according to the parametrization by Zuo

and Oerlemans (1996). This leads to the gradual decrease of water content within the aquifer. The GEUS model incorporates

a Darcy-like parametrization of the subsurface runoff, which results in faster drainage of the aquifer than the Zuo-Oerlemans parameterization. However, observations showed that excess water in the aquifer does not run off immediately but flows laterally and can remain in the aquifer for several decades (Miller et al., 2019).

Another challenging question for understanding and modelling of firn aquifers is: Where and when does the meltwater generated at the surface percolate down to the aquifer? Firn temperature observations show that the top 20 m of firn remained at melting point during the 2014 melt season. This indicates that meltwater from the surface reached the aquifer. The firn models do not conclusively answer how and where deep percolation to the firn aquifer takes place. Given the same surface forcing and initial firn conditions, only the models with explicit deep-percolation schemes (CFM-Cr, CFM-KM and

UppsalaUniDeepPerc) infiltrate water down to the aquifer. This could indicate that the recharge of the firn aquifer has to be through heterogeneous percolation because it is the only way firn models can mimic observations. However, such a systematic infiltration through vertical channels should leave visible traces in the form of altered stratigraphy, ice columns (Marsh and Woo, 1984) or show repeatedly in firn temperature observations when meltwater infiltrates into cold firn in spring (Pfeffer and Humphrey, 1996; Charalampidis et al,2016). Future field investigation should ascertain whether

preferential flow is indeed the only process infiltrating water to the aquifer. Another interpretation could be that models using a bucket scheme (DTU, IMAUFDM and UppsalaUniBucket) or Darcy's law (DMIHH, GEUS and MeyerHewitt) do not infiltrate water deep enough because of inappropriate irreducible water content or firn permeability for the firn aquifer site. Yet, few in situ datasets are available to constrain these firn characteristics in the models. One last possibility could also be the misrepresentation of surface conditions: the melt calculated at the surface is subject to the biases and the uncertainties

that apply to the so-called "bulk approach" used here in the energy budget calculation (Box and Steffen, 2001; Fausto et al., 2016). Although it was ensured that the calculated skin surface temperature agreed with observations available at KAN_U and FA, no direct observation of melt is available at our sites. Furthermore, the horizontal mobility of the meltwater, especially at high-melt sites such as FA, could lead to the injection at the surface of more meltwater than what is being melted. Therefore, more work is needed to quantify liquid water input at the top of the model in the firn aquifer region.

**6.   Towards ensemble-based uncertainty estimates for firn model outputs**

Given the complexity of the firn models, it is difficult to propagate uncertainty and account for model assumptions and parameterizations. As a consequence, firn model outputs have commonly been given without uncertainty range which prevents assessing the robustness of model-based inferences. Taking inspiration from previous ensemble-based modelling approaches (e.g. Nowicki et al., 2016), we provide a multi-model estimation of the uncertainty that applies to any simulated

value of firn temperature and density, and more importantly, to the simulated values of meltwater retention (through refreezing) and runoff.

**6.1.  Firn temperature and density uncertainty**

We see from Figures 2 to 7 that the spread among models increases as we move from the dry snow area to the percolation area, peaking in areas with high-melt features such as ice slabs and firn aquifers. We suggest that the model spread presented here can provide a baseline for uncertainty whenever a single model is used. At Summit, representative of the dry snow area, modelled average densities in the top meter of firn have a standard deviation of 13 kg m$^{-3}$. Hence, a two-standard-deviations ($\pm2\sigma$) uncertainty envelope of $\pm26$ kg m$^{-3}$, or $\pm8\%$, can be used to describe the modelling uncertainty. At Dye-2, representative of the percolation area, the top 1 m average density simulated by the models have a maximum standard deviation of 145 kg m$^{-3}$ during the 15-years-long simulation. This indicates that a substantial level of uncertainty, $\pm290$ kg m$^{-3}$, or $\pm75\%$, applies to the modelled average density for the top meter. Similar uncertainty ($\pm77\%$) applies to the modelled top 1 m average density at KAN_U. As for density, the model spread in simulated firn temperature can be investigated by calculating the maximum standard deviation of firn temperature at 5 m depth among models. At Summit the $\pm2\sigma$ uncertainty envelope on simulated 5 m firn temperature is $\pm4$°C. This model uncertainty envelope is wider at Dye-2, $\pm14$°C, because of the different meltwater infiltration depths simulated by the models. At KAN_U, the uncertainty in 5 m temperature, within the ice slab is $\pm10$°C. The uncertainty range increases closer to the surface and at sites or depths where meltwater infiltration may be captured differently by the models. The level of uncertainty, both for density and temperature, increases when narrowing the depth range over which averages are calculated, and conversely. This result indicates that firn models are still very variable when considering a specific depth but agree better when looking at the average firn property over a larger depth range. The uncertainty ranges provided here represent the largest deviation seen among models at any three-hourly time step and are therefore conservative. They can nevertheless be used as a metric for uncertainty in the absence of observational constraints or when using a single model.

### 6.2. Mass balance

The differences among simulated firn density, temperature, and liquid water distribution can cause them to retain and run off different amounts of meltwater and therefore affect the surface mass balance. The models agree that all meltwater is retained at Summit and Dye-2_16. At Dye-2_long and KAN_U, the inter-model average and $\pm2\sigma$ values can be used as a multi-model estimation of the meltwater retention, runoff and of the uncertainty on these estimates.

At Dye-2, the DTU model produces unrealistic runoff values (Figure 11c) because of the impermeability of near-surface ice layers blocking downward percolation and enhancing runoff. This highlights how a model designed for the dry snow area (Simonsen et al., 2013) can fail to capture meltwater retention in the percolation area. We therefore do not consider this model in our multi-model uncertainty estimation. All the other models agree that runoff is minimal compared to refreezing at Dye-2 (Figure 11a-c). CFM-KM and CFM-Cr are the only models that calculate minor runoff some of the years (Figure 11b). This is likely linked to the buildup of denser firn layers close to the surface (Figure 4) through which water in the matrix flow domain could not percolate. Even though the preferential flow domain could infiltrate some of the meltwater at depth (Figure 4c) this was insufficient to accommodate all the meltwater input. As a consequence, in 2012, year with the

highest meltwater input, models on average calculate that $27 \pm 119$ mm w.e. is run off, $3 \pm 13\%$ of the meltwater input (Figure 11 b, c). The large uncertainty envelope applying to calculated runoff highlights the disagreement of models during high melt years (Figure 11b). In years with absent or minor runoff, the annual refreezing totals reflect the surface melt prescribed to all models (Figure 11a).

At KAN_U, the impact of the ice slab on the surface mass balance is critical. The different simulated meltwater infiltration patterns (Figure 8c) lead to varying total amounts of meltwater either refrozen or runoff (Figure 11a-c). The bucket schemes (IMAUFDM, UppsalaUniBucket) and UppsalaUniDeep percolate meltwater through the ice slab and refreeze all of the input meltwater. In all the other models, the presence of ice layers prevents or slows down meltwater infiltration, triggers ponding and lateral runoff, including in the CFM models where the preferential flow domain is unable to accommodate all the incoming water. The lowest melt year, 2015, has the lowest model spread with $304 \pm 80$ mm w.e. of the meltwater refrozen, $97 \pm 17$ % of the total meltwater input (Figure 11). The highest melt year, 2012, also has the highest model spread in annual refreezing with $913 \pm 557$ mm w.e. of water refrozen, $73 \pm 48$ % of the meltwater input (Figure 11). Subsequently, the average runoff among models in 2012 is $353 \pm 610$ mm w.e., about $27 \pm 48$ % of the prescribed surface meltwater (Figure 11). For comparison, Machguth et al. (2016) calculated from firn cores that $75 \pm 15\%$ of the surface meltwater ran off at KAN_U in 2012. Although the observations are subject to considerable uncertainty, they indicate that most of the models underestimate the runoff at KAN_U in 2012. Yet, the uncertainty envelope that apply to the simulated runoff in 2012 includes both zero runoff and the observed value (Figure 11f).

We do not evaluate meltwater retention and runoff at FA owing to the major limitations that we highlighted in the current handling of firn aquifers in firn models. Indeed, modelled runoff, traditionally defined as excess water entering an efficient drainage system and leaving the ice sheet, does not occur at FA (Miller et al., 2018). Instead, the excess water saturates the firn and slowly moves downstream within the aquifer, that none of the models can represent. In the percolation sites represented here by Dye-2 and KAN_U, the model spread generally increases with increasing surface melt and when more of that meltwater runs off (Figure 11). This inter-model variability largely stems from the differences in meltwater infiltration and refreezing patterns which themselves depend on meltwater input (see Sections 4.2, 4.3 5.2 and 5.3). We therefore highlight the disagreement of the firn models in their simulations of the meltwater retention, refreezing, and runoff in the lower accumulation area of the ice sheet. High-melt accumulation areas should therefore be the subject of further field investigations to ascertain the actual meltwater retention there and better constrain firn models.

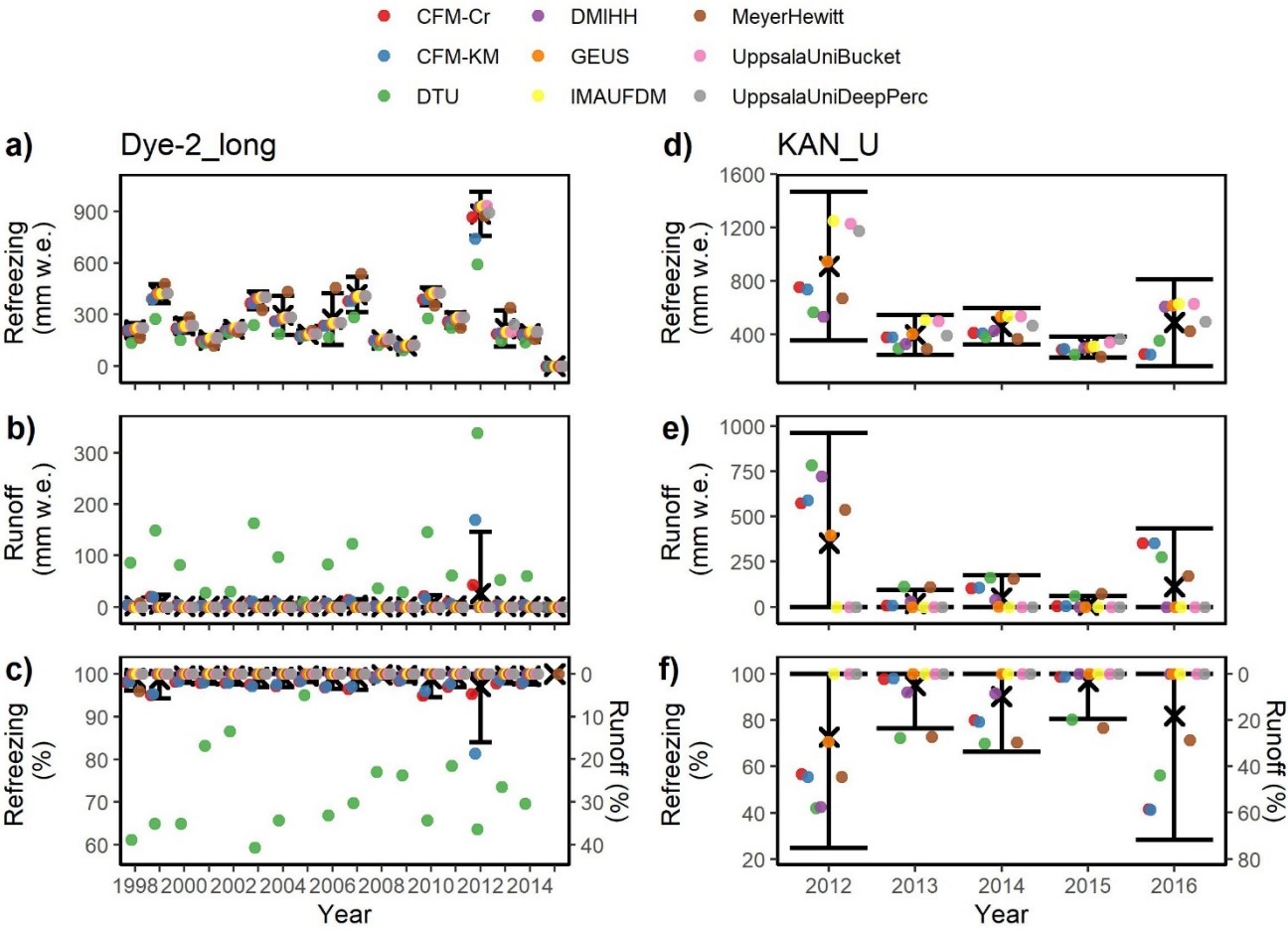

**Figure 11. Yearly meltwater refreezing (a,d) or runoff (b,e), as totals (a,b,d,e) or fractions of the total meltwater input (c,f) at Dye-2 (a,b,c) and KAN_U (d,e,f). For each panel, yearly inter-model averages (black cross) and ±2σ values (error bars) are calculated from all models except the DTU model.**

## 7. Summary remarks and perspectives

Nine state-of-the-art firn models were forced with mass and energy fluxes calculated from weather station data at four sites representative of various climatic zones of the Greenland ice sheet. From the intercomparison of their simulated firn temperature, density, and water content, and from evaluation against various firn observations, we identified specific routines within the models that are responsible for the models' behaviors. We later quantified uncertainties that apply to the firn model outputs and on their evaluation of meltwater retention. We identified key topics for future development of models and for the investigation of firn processes.

We identified the following disagreements among models and discrepancies between model outputs and observations. Runoff-enhancing ice slabs were formed in certain models at the Dye-2 site where they are not observed. At the KAN_U site, where models were initialized with a several-meter thick ice slab according to observations, models do not agree whether such ice layers allow meltwater infiltration or not. Models that explicitly include deep percolation allow water infiltration through the ice slab, which is incompatible with the relatively cold firn observed at depth. At the aquifer site, only deep-percolation models infiltrate meltwater to the aquifer. Nevertheless, all models misrepresent the aquifer either because of the inability of some models to simulate saturated conditions, the different time scales at which the excess water is sent to runoff, and the absence of horizontal subsurface water movement. At all sites, Eulerian models smooth the firn density profile and dissipate contrasts in firn density even in a model with high vertical resolution. Model spread and deviation between simulated and observed firn density and temperature is largest at the sites that experience more melt. Using twice the standard deviation in model outputs as an indicator of uncertainty envelope, we found that firn models can estimate firn density within $\pm60$ kg m$^{-3}$ at a dry snow site and that uncertainty increases to $\pm290$ kg m$^{-3}$ for certain depth ranges at percolation sites. The similarity between modelled and observed firn density at the nearly melt-free Summit site indicates that for the top 20 m of firn, the densification equations perform similarly under dry-snow conditions given identical forcing. However, variability in simulated firn temperature at Summit indicates that heat transfer through the firn is still not handled consistently in firn models. Consequently, none of the tested models compared positively with observations at all four sites.

Differences in simulated firn characteristics in the nine models led to different amounts of meltwater being retained through refreezing or being lost through runoff. Models that percolate meltwater deeper (resp. shallower) calculate higher (resp. lower) retention through refreezing and therefore less (resp. more) lateral runoff. The spread among models regarding annual meltwater retention is positively correlated with surface meltwater input and is maximal, on absolute values, at KAN_U in 2012, the highest melt year. Still, during that year, the inter-model average runoff is only $27 \pm 48\%$ of the total meltwater input. Therefore, further work is needed to evaluate firn models where or when even a higher fraction of the input meltwater runs off.

These mixed results show that even the newest models need further development to perform satisfactorily under the wide range of climate and firn conditions of the Greenland ice sheet. We recommend the following topics for future investigations:
- More observations of firn permeability should be conducted both at point and regional scale. Measurements of grain size and other microstructural properties would also help to evaluate the parametrizations currently used by some of the firn models for permeability. These measurements should focus on the lower percolation area where meltwater infiltration and runoff play an important role in the surface mass balance.
- Bucket schemes, which do not calculate firn permeability, would benefit from a density-based impermeability criterion. This criterion needs to be drawn from field evidence at the scale at which the models operate.
- Recent work on firn thermal conductivity (e.g. Calonne et al., 2019, Marchenko et al., 2019) should also be used to improve the firn models. Furthermore, the impact of vertical ice features and firn ventilation on firn temperature is

 currently not included in any of the firn models. Firn temperature observations are now available to assess model performance and should be part of the standard evaluation protocol.

- Eulerian models should be used bearing in mind that they gradually smooth firn characteristics. This issue does not prevent the use of such models, as long as the features that are being studied (e.g. ice slab, runoff, firn aquifer...) are being defined in ways that are compatible with the Eulerian framework.

- A major rethinking of firn models is necessary to better represent firn aquifers. In these regions, models need to allow saturated conditions and lateral subsurface water flow either explicitly with a multi-dimensional model or through an adapted parameterization. More field observations are also needed to ascertain the surface meltwater input at these sites, whether near-surface drainage occurs and, if it does, the size of such drainage area.

- Recent efforts were made to explicitly describe heterogeneous meltwater infiltration in firn models. While they allowed better performance at the firn aquifer site, they infiltrate water too deeply and produce positive biases in firn temperature at the dry snow site and the two percolation sites. Further work is needed to understand, under various surface and firn conditions, when heterogeneous percolation occurs, how deep it should reach and how much water it should transport. Only after these questions are understood can a reliable preferential flow scheme be developed.

- The fresh snow density is known to have an impact on the firn model outputs but was here set to a site-invariant value derived from observations. Fresh snow density is known to vary considerably in space and time although no statistically robust parameterization exists up to date for the Greenland ice sheet. Future measurement campaigns and modelling efforts could help to prescribe surface snow density and to better capture its interactions with the densification and heat transfer scheme.

Considering the number of firn characteristics that remain to be investigated and the cost of field surveys, laboratory experiments could be highly valuable if they can address the boundary effects, the scale of the process being investigated, and provide realistic surface and firn conditions. Investigation of the points listed above will collectively improve our understanding of firn and meltwater dynamics, improve the representation of these processes in firn models, and eventually reduce the uncertainty that applies to their output when assessing the surface mass balance of the Greenland ice sheet in past, present, and future times.

## 8. Code and data availability

The scripts and datasets produced for this study are available at the following links:

- Surface forcing data: https://doi.org/10.22008/FK2/GZ3CSN (Vandecrux, 2020)
- RetMIP protocol and metadata: http://retain.geus.dk/index.php/retmip/
- Model outputs: https://doi.org/10.22008/FK2/CVPUJL (Vandecrux et al., 2020b)
- Plotting scripts: https://github.com/BaptisteVandecrux/RetMIP
- CFM model code: https://github.com/UWGlaciology/CommunityFirnModel

- GEUS model code: at https://github.com/BaptisteVandecrux/SEB_Firn_model

## 9. Funding

This work is part of the Retain project funded by the Danish Council for Independent Research (Grant no. 4002-00234) and the Programme for Monitoring of the Greenland Ice Sheet (www.PROMICE.dk). Achim Heilig was supported by DFG grant HE 7501/1-1, Horst Machguth acknowledges support by ERC CoG Nr. 818994 The AWS used at Dye-2 during the 2016 melt season is supported by the Natural Sciences and Engineering Research Council (NSERC) of Canada. ArcTrain and Arctic Institute of North America (NSTP). Olivia Miller and Clifford Voss were supported by the U.S. Geological Survey. C. Max Stevens and Michael MacFerrin were supported by the National Aeronautics and Space Administration (NASA) grant NNX15AC62G.

## 10. Author contribution

RM, PL, RF and JB secured and administrated the funding, conceptualized and supervised the RetMIP project. PL, MS, VV, SL, PKM, SM, WvP CM, SS and BV provided the model runs. AH, SS, SM, HM, MM and BV provided the observations against which the models could be evaluated. MO participated to the data visualization. BV, with input from the co-authors, designed the methods, conducted the intercomparison and wrote the original draft. All co-authors contributed to the review and the editing of the manuscript.

## 11. Acknowledgement

We are grateful to Ian Hewitt for his insight on the MeyerHewitt model. We thank our scientific editor Xavier Fettweis as well as Samuel Morin, Kendall FitzGerald and an anonymous reviewer for comments and suggestions that significantly improved the study.

## 12. Competing interests

The authors declare that they have no conflict of interest.

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
