# Peer review of "The firn meltwater Retention Model Intercomparison Project (RetMIP): Evaluation of nine firn models at four weather station sites on the Greenland ice sheet"

_The Cryosphere, 2019_

## Referee Comment (RC1) · Anonymous Referee #1 · 14 Apr 2020

**Review:** *The firn meltwater Retention Model Intercomparison Project (RetMIP): Evaluation of nine firn models at four weather station sites on the Greenland ice sheet*, Vandecrux, et al.

**Summary**

In *The firn meltwater Retention Model Intercomparison Project (RetMIP): Evaluation of nine firn models at four weather station sites on the Greenland ice sheet*, Vandecrux et al. present an assessment of nine state of the art firn meltwater retention models, forced by data from four sites spanning a range of conditions in Greenland's percolation and dry snow zones. The authors find that densification schemes perform similarly, as evidenced by limited model spread at Summit station, where melt events are rare. In contrast, at sites in Greenland's percolation zone, where seasonal melting is commonplace, models simulate a wide range of density and temperature, which is supposedly primarily influenced by different infiltration schemes between models. Based upon the results, a host of discussion points are raised indicating shortcomings of particular models that need to be addressed.

The fate of pore space in Greenland's firn package has important mass balance implications for the ice sheet. The runoff/retention of surface melt depends on the detailed physics of meltwater infiltration, which remains a central challenge for the field (van As, 2016). While this manuscript does not present any fundamental scientific advancements towards overcoming this challenge, the intercomparison is an opportunity to synthesize the present state of the modeling field and range of output from various published models. This, I believe, has merit and is worthy of publication.

The topic is certainly of high relevance, but I believe the manuscript requires major revisions before it can be considered for acceptance. In its present state, the manuscript fails to present model output which is most relevant to the community. This could be alleviated by refocusing the results section with greater attention to the details of simulated infiltration between the different models. Additionally, I found the Discussion to be quite unorganized, reading largely as a scattered list of model details, and introducing new results. Reorganizing the Discussion, detailing how different infiltration schemes lead to wide differences in modeled density and temperature, and providing better synthesis of the model results with the objective of outlining directions for improvement, would substantially improve the manuscript.

These primary concerns are detailed below. In addition, I have included specific comments where language, content, and figure edits would improve the manuscript.

**Major concerns**

*Presentation of results*

As the title of the paper states, the manuscript presents a comparison of firn models with explicit schemes for accommodating the infiltration, refreezing, and runoff of surface melt. This distinguishes the study from past firn model intercomparisons (e.g. Lundin et al (2017)), and compels the authors to present intercomparison results that leverage the unique capabilities of the models. Specifically, I believe the manuscript would be of much greater interest to the community if the authors present meltwater infiltration and refreezing results explicitly. In its current form, the intercomparison results are focused predominantly on density and temperature. Minor focus is given to infiltration, refreezing, and retention, and the reader is forced to interpret the differences in infiltration/refreezing among the models through the density and temperature results.

The depth of infiltration at the different sites/years, and the annual depth distribution of refreezing are quantities that would be very useful in presenting the range of simulated meltwater infiltration among the models. I recognize there are few observational validation metrics, but the authors do have the upward-looking radar results (I found this figure to be the most informative in the manuscript). But even so, one of the objectives of the manuscript I believe is to communicate the range of meltwater infiltration in the retention models that are out in the community. Since infiltration/refreezing is also of first order importance in controlling the simulated density and temperature range, a greater focus on these quantities provides a much richer context for discussing the density and temperature results. This is especially true considering that observations of these variables are more common.

*Organization of discussion*
Much of the discussion reads as a scattered list of interesting things the authors noticed in their results, rather than a synthesis of the model results with direction for the community. This is especially true of section 5.1. I think the authors have an opportunity to use detailed infiltration results (see above) as a foundation to discuss why density and temperature are so variable. The Summit density results show that the underlying densification schemes in the models are similar, even when temperatures span a wide range. So the large range in density results at percolation zone sites is due primarily to differences in infiltration. This point would be strengthened if the authors present the refreezing results. By framing the density/temperature output within the context of infiltration, and comparing these (density/temperature) fields to observations, the authors can expand on how the general infiltration approaches perform and comment on directions for improvement. This approach seems more logical than the scattered listing of 10 details and improvements. It may also feed more naturally in to the discussion of density and temperature uncertainty in models.

Section 5.2 focuses on uncertainty in density, temperature, and mass balance from the suite of model output. I found the mass balance uncertainty particularly interesting, but I also consider these to be results, and should be presented as such in section 4 (Results). The Discussion section here is a unique opportunity to discuss the implications of the infiltration schemes for quantities that are of perhaps the most importance to many -- runoff. The authors offer discussion of specific sites but any sort of upscaling is lacking. Moreover, the results from Dye-2 are used to predict future conditions at the site, which I believe is both a non-sequitor and not necessarily supported by the results. Based on the presented results from KAN_U it is apparent that the different infiltration schemes result in a huge range in runoff/refreezing from model to model. If the Machguth (2016) results are to be accepted, then it suggests that perhaps the retention models underestimate runoff. However, Machguth et al. (2016) also define Dye-2 as a deep percolation site outside the runoff regime. Yet, the DTU model, which shows the highest runoff at KAN_U, also indicates that 30-40% of surface melt runs off at Dye-2. Based on: a) the wide spread in modeled runoff, and b) the fact that the model which supposedly most closely honors the interpreted runoff at KAN_U also appears to overpredict runoff at Dye-2, my interpretation is that the current generation of infiltration/refreezing models do not have the capacity to quantify runoff with fidelity. Is that an accurate assessment? If so, this would be a very important point to make to the community.

I struggled with section 5.3. It outlines many shortcomings of the models' ability to treat firn aquifers but does not draw explicitly from anything in the Results. The Results section simply states that the case of the firn aquifer is discussed in 5.3 (line 312). If the authors are to discuss model shortcomings, then they need to at least support these criticisms with presented results. Restructuring the results section by site (presenting infiltration, density and temperature site by site) would be one way to help achieve this. This way, for instance, the reader will be primed on whether models simulate firn reaching the melting

point down to 20m as is shown in observations. This appears to occur in a number of models in Figure 5 but the text in the Discussion implies that this may not be the case.

**Specific comments**
ln 28: change '...capacity to retain part of the surface meltwater...' to '...capacity to retain part or all of the surface meltwater...'

Abstract: The abstract contains only methods and results. No synthesis of interpretations or summary statement.

ln 44: Consider replacing 'seen' here and elsewhere in the manuscript. 'occurred'?

ln 45: See comment re: ln 44

ln 60-65: In the first para, the importance of firn for meltwater retention and observed changes are established. In this second paragraph it would be stronger to communicate the challenges associated with meltwater infiltration in firn. Reijmer presented early work modeling infiltration/refreezing. But new models are on line with differing and, in some instances, more physically based schemes for meltwater infiltration/refreezing. Others have also tested different melt infiltration schemes. What about Steger (2017)? The authors need to do a better job of placing this work in the context of existing comparisons of meltwater infiltration/retention.

ln 66: Why just 'some' of the models currently used on the GrIS? What other models are the authors explicitly *not* using, and why is this the case?

ln 70-73: I would consider reframing the questions to more explicitly treat the different infiltration schemes. Isn't this the point of the study, to compare the infiltration schemes of different model schemes and the impact on the firn framework (density and temperature), and resulting runoff/refreezing partitioning?

ln 117-119: Reeh (2005) limits melt to the annual layer while the bucket approach redistributes based on cold and liquid water content. Seems to be contradictory?

ln 124-125: Is this statement relevant to the output? It appears to be a technical detail of the model mechanics that is a non-sequitor.

ln 141: Vandecrux (in review) does not appear in the references?

Table 2: GEUS model, Runoff Calculation -- change 'identical' to 'adjacent'? Also, DMIHH Meltwater routing is empty. Should this also be Darcy's law?

ln 210: The authors state that 'as many boundary conditions as possible' were given to all modes. Is there any guarantee that the models are all being forced by the same surface melt? This is imperative to understanding the results. Hopefully the answer is 'yes'. I believe, based on reading of the supplemental, that this the case for all but the Meyer and Hewitt model? I think this should be stated explicitly in the manuscript. Moreover, the authors should state in the main manuscript an estimate of the high/low melt bias in the Meyer and Hewitt model.

Figure 4: With such narrow panels this plot is impossible to decipher. Are models grouped and cores repeated 3x for clarity? What about 2 plots for each core? 1) the measured core density, and 2) the difference between measured and modeled for each model?

Figure 6: Again, the plot is indecipherable. All the text overlaying the plots implies that the visuals themselves are unimportant. If so, then just present the ME, RMSE, and R2 in a table and be done with it.

ln 303: 'more' is very qualitative. The results also appear to be very model-dependent at KAN-U. So what is 'more'? Consider just eliminating it.

ln 319-321: I found this lead-in confusing b/c the next figure call (Figure 7) references is all sites. Further, the next Figure 8, is focused on Dye. As stated above, I would consider restructuring the results by site.

Figure 8: State in the caption that a, b, c are all the same. Just grouping models for clarity.

ln 339-340: awkward sentence

ln 369: what is 'dissipate contrast'?

ln 382: What is inappropriate? Nearly all of the models use a thermal conductivity referenced in the literature.

ln 384: Incomplete sentence.

ln 407-409: Interestingly, Charlampidis et al. (2015) show what they believe to be refreezing below this very same ice slab. Just delayed in time.

ln 414-415: This could then indicate the challenge of comparing a firn model, which operates over such large spatial scales, with measurements of density that are local and prone to large spatial variability.

ln 526-527: These are new results in the conclusion!

ln 532: The paragraph leading to this statement summarizes what I believe to be the primary purpose of the manuscript. Synthesis of the model results beyond site-by-site agreement would vastly improve the paper. The final sentence is also incomplete.

Supplemental: Considering the revisions required in the manuscript, I have not gone through the supplemental material in detail. However, figure call-outs do not follow a consistent order (e.g. Figure ST1 is followed by Figure S1).

---

## Short Comment (SC1) · 28 Apr 2020

This report is significant in that it compares the performance of multiple firn models in Greenland, run using the same forcing data and boundary conditions, to field observations. The importance of this work, however, is obscured by the poor organization of the report, particularly in the discussion section. Improvements to the organization of the report through the implementation of an interpretation framework and systematic discussion would increase its relevance to the modeling community.

[Figure]

The discussion currently reads as a list of findings, instead of taking the reader through the results in a systematic way. Organizing the discussion using a framework (e.g. in terms of model type, site, result type) similar to that used in the results section would improve clarity. Additionally, it is unclear why some sections are included in the discussion section. For example, section 5.1.10 discusses the importance of the value used for fresh snow density in the model; however, this section cites only previous studies and does not discuss how this is demonstrated by the results of this work. Additionally, with the many abbreviations used for model and site names, keeping track of the properties of each model without constant reference to Table 2 is difficult if the reader is not familiar with all the models. Having a framework in which these models are referenced, and a more systematic discussion of results, would improve readability.

This report is of particular interest to those deciding between firn models to use and those interpreting results from such models. As such, the results of this paper could be presented in a way to highlight the effect that these findings have on choosing or interpreting these models. These findings are touched on in the abstract but are difficult to parse from the discussion. Better organization of the discussion, as previously suggested, would remedy this issue.

---

## Referee Comment (RC2) · Samuel Morin (Referee) · 7 May 2020

The manuscript by Vandecrux et al., entitled "The firn meltwater Retention Model Intercomparison Project (RetMIP): Evaluation of nine firn models at four weather station sites on the Greenland ice sheet", provides results from a recent intercomparison of firn models dedicated to their handling of meltwater retention and runoff. This topic has attracted a lot of interest over the past decade, following the discovery of sub-surface aquifers on the Greenland ice sheet and the fact, more generally, that liquid water transfer is a critical, yet very complex process governing the energy and mass

balance of firn below the surface. The manuscript undoubtedly fits the scope of The Cryosphere. It is generally well designed and easy to understand, although, like many other manuscripts reporting on intercomparisons, the reading can be a bit cumbersome and the results will mostly appeal to experts in this field. I recommend publication of this article, and I provide here below some feedback and suggestions, which may be considered by the authors in case they lead to improvements of the manuscript.

Page 1, line 1-2: "Perennial snow, or firn, covers 80% of the Greenland ice sheet and has the capacity to retain part of the surface meltwater, buffering the ice sheet's contribution to sea level". This sentence could in fact be quite misleading and I suggest reformulating, either at least acknowledging that the "buffering" acts upon sea level *rise*, or maybe more generally, leave out the term "buffering" (which could be inappropriate in some cases where runoff generation can increase the contribution of the ice sheet to sea level rise), and use a more neutral term referring to the fact that firn processes influence the behaviour of the ice sheet and affect its contribution to sea level rise.

Page 1, line 3 : "weather-station-derived" : maybe consider unpacking the wording, this is quite tedious to read.

Page 2, line 60 : I think it could be useful, and appeal to a wider community of readers, if this article could provide more information on what a "firn" model is, or what a "snow and firn" model is if the two terms here are meant to be combined. Indeed, GCMs and RCMs all feature a "snow" component in their land surface models, and many such models do not handle ice sheets differently from a long-lasting seasonal snowpack. It could thus be quite appropriate to provide more background on how snow and firn processes are considered in GCMs and RCMs (or other tools used for reanalyses etc.), in order to make this effort even more useful in a CMIP6/IPCC context. This could make it possible to establish how the firn models used for this intercomparison fit in this wider context, thereby providing information on whether the results of this intercomparison are relevant when discussing the results of existing GCMs and RCMs projection. Also

(see below point too), it would be good to introduce what are the typical input/output to firn models, because they seem to deviate from typical input/output of land surface models, hence the link to GCMs/RCMs/etc. is not direct.

Page 2, line 64 : Although I know this can be debated, I have strong personal reservations against the term "validation", which I believe is beyond reach in any geosciences field because the "truth" is never known hence "validation" (in the strong sense) can only be elusive. I much prefer the term "evaluation", which better encapsulates the fact that the evaluation results from a comparison with observations, which also carry some uncertainties. I was happy to see the term "evaluation" in the title and in the abstract, so it was disappointing to see the term "validation" popping up in the introduction. Maybe this can be harmonized throughout. Along the same line, in several instances the comparison between model output and observations is referred as a "bias". Here again, "bias" is a term which includes some judgement of value, implicitly assuming that observations are the "truth". I think that observations are never the "truth" and, especially regarding in-situ snow and firn observations, we know that observations are intrinsically prone to significant errors, in addition to large spatial variability for many variables at all scales, which induces representativeness issues. In this context, I much prefer referring to "deviations" between model results and observations, without using the term bias. I note that the term "deviation" is used in several places in the manuscript, including in figure captions (e.g. caption of Figure 6, which I think is perfectly worded), so maybe this could be harmonized in the text.

Page 2, line 67 : Here is a good example where more information could be provided on whether (some of) the firn models included in this intercomparison are representative to how firn processes are handled in GCM/RCM/NWP models, so that the results can be used to analyse some of the output of such models. At present, and even though some of the information is provided in section 2, the models included in the intercomparison are not categorized explicitly according to their use, and I think it could be helpful to the scientific community to provide the rationale and the results of the intercomparison in

a way that can (somehow) be transferred to the interpretation of other model results.

Page 3, line 91 : it seems that there is a typo in this reference, I believe this should be 1998. I haven't checked all reference, but if they were typed by hand and not using literature management software, there may be other errors in the references.

Page 4, line 1 : This table is very useful, I suggest adding a column for providing the extended name (developing the acronym) and, more to the point, adding a column on how the model is typically used (included/coupled to a land surface model in RCM/GCM/NPW context, or purely offline for process investigations etc.). I'm convinced that the authors can easily define several categories within which models can be classified (these categories could even be used in the results and discussion, if common features, or not, emerge from these various categories, in addition to discussing results referring to process representation in models).

Page 4, Line 98 : I think it would be good to spell out the acronyms in the titles of subsections 2.1, 2.2 etc., this is otherwise quite obscure for non-expert readers not accustomed to the acronyms of these firn models.

Page 5, line 116 : extra "-" between ASIRAS and instrument

Page 5, line 131 : it seems to me that layers defined by a w.e. (e.g. mass per unit surface area) should not be qualified by a "thickness", which refers to a distance (in m). If the model is formulated in terms of layers with a given mass, then I believe the text should refer to this, and the substitution of "thickness" with "mass" will accurately represent how the model is formulated.

Page 7, Table 2 : I think that the column on "Hydraulic conductivity" needs some attention. The van Genuchten (1980) article provides a way to link between the saturated hydraulic conductivity and the hydraulic conductivity, which for snow has been addressed by several studies such as Shimizu (1970), Calonne et al. (2014) or can be used using geometrical estimates such as Carman-Kozeny (see Calonne et al., 2014

for a review of existing formulations, and Wever et al., 2014, for context). Hence I suggest to double check, for each model, what is the parameterization used for estimating the saturated hydraulic conductivity (corresponding to the permeability) from the microstructure (density, specific surface area/grain size) and the formulation used to derive the hydraulic conductivity (van Genuchten (1980) is probably widely used). This column seems to be lumping and mixing the two.

Page 8, line 177 : I'm not fully convinced by the formulation of how the forcing data are introduced. "Any bias in forcing data propagates into the model output" : I'd rather suggest that any \*difference\* in forcing data propagates into differences in model outputs. The term "bias" is here inappropriate I think (see comments above). Further, "To make sure we compare and evaluate the models independently of biases that may exist in forcing datasets that come from RCMs, we use meteorological fields derived from five weather stations at four sites." I think this sentence needs rephrasing, because it gives the impression that only RCM atmospheric fields can be biases, and in-situ atmospheric observations are not biased. I don't see the point in referred to RCM here at all, but simply state that "To make sure we compare and evaluate the models independently of differences due to forcing data, we use for all models the same meteorological fields derived from five weather stations at four sites." The references to RCM data is absolutely not needed here, and in the current formulations I consider it misleading and improper.

Page 9, line 202 : If I understand well, the firn models are driven by 3-hourly skin temperature, meltwater generation (what it this ?) and net snow accumulation. I think this warrants an explicit statement on the forcing data for firn models (see my comment regarding the introduction), because this appears to be quite different from forcing data of land surface models (including the snow component), usually driven by air temperature, relative humidity, incoming shortwave radiation, incoming longwave radiation, wind speed (and direction) and snowfall and rainfall rate (in offline or online applications). In this context, it would be good to quickly introduce how the firn model

data input are typically computed within GCM/RCM/NWP models where they are implemented. I also think that it would be beneficial for the manuscript, if the material provided in the Supplement regarding how the models were adapted in order to contribute to the intercomparison, could be placed within the body of the article. [In fact, I'm in favour of moving the whole supplementary material into the article, which is quite technical anyway, and which I think would benefit from having all the information in the same document].

Page 10, Table 3 : Mean annual air temperature data should probably be homogenized in terms of formatting (why is the number provided by 0.1°C resolution of KAN_U and rounded to the unit °C for other sites ?)

Page 20, line 371 : Eulerian

Page 20, line 384 : it seems that the sentence does not end as planned. Was it the intent to refer to Calonne et al. (2019 ; https://doi.org/10.1029/2019GL085228 ) which may provide some hints into how to parameterize firn thermal conductivity ?

Page 21, line 404 : The last sentence "Especially, the heterogeneous nature of the firn, the presence of vertical ice features in the firn, the variability in surface snow density/thermal conductivity as well as firn ventilation are processes not currently included in the models and should be subject of future research." is certainly true but is quite vague. Based on existing literature (e.g. Albert et al.) is it possible to elaborate more on the expected direction of change and whether accounting for such processes would be beneficial (and at what numerical cost) ?

Page 21, line 413 : add "with" after "match"

Page 22, line 433 : on the "fresh snow" density issues and beyond, there is little discussion in the manuscript on the connection between snow cover models (such as those embedded in land surface schemes) and firn models. Wouldn't it be adequate that surface processes are handlded by snow cover model rather than firn models, taking

advantage of the features of both types of models ? This connects with the question on how firn models are used within coupled GCM/RCM or offline (see comments anove).

Page 22, line 443 : It should be mentioned here that the uncertainty envelope corresponds to +/- 2 standard deviation (+/- 2 sigma). This information is missing here (although it is provided later in the conclusion). Regarding the temperature, I see a limitation to the estimate provided here, in the sense that the temperature of the firn cannot exceed 0°C. How does this impact the uncertainty range provided for temperature ?

Page 23, line 451 : What is the argument for stating that this uncertainty range applies in situations where observations are not available ?

Page 24, line 482 : "much faster drainage of the aquifer" : faster than what ?

Page 24, line 484 : A qualifier is probably missing before "models" : "existing firn models" ? "firn models considered in this intercomparison" ?

Page 24, line 490 : I suggest not using the term "reality" in scientific publications. Furthermore, a reference is missing to support the statement in this sentence.

Figures : I didn't notice any major flaw in the design or content of the figures, which are appropriate to convey the results of this intercomparison.

---

## Author Comment (AC1) · 10 Jul 2020

**Response to the reviewer 1**

Dear colleague,

We are grateful for your constructive review of our manuscript. Please see our response (in green text) to each of your comments and suggestions below.

Sincerely,

Baptiste Vandecrux on behalf of the co-authors.
* * *
Review: The firn meltwater Retention Model Intercomparison Project (RetMIP): Evaluation of nine firn models at four weather station sites on the Greenland ice sheet, Vandecrux, et al.

Summary

In The firn meltwater Retention Model Intercomparison Project (RetMIP): Evaluation of nine firn models at four weather station sites on the Greenland ice sheet, Vandecrux et al. present an assessment of nine state of the art firn meltwater retention models, forced by data from four sites spanning a range of conditions in Greenland's percolation and dry snow zones. The authors find that densification schemes perform similarly, as evidenced by limited model spread at Summit station, where melt events are rare. In contrast, at sites in Greenland's percolation zone, where seasonal melting is commonplace, models simulate a wide range of density and temperature, which is supposedly primarily influenced by different infiltration schemes between models. Based upon the results, a host of discussion points are raised indicating shortcomings of particular models that need to be addressed.

The fate of pore space in Greenland's firn package has important mass balance implications for the ice sheet. The runoff/retention of surface melt depends on the detailed physics of meltwater infiltration, which remains a central challenge for the field (van As, 2016). While this manuscript does not present any fundamental scientific advancements towards overcoming this challenge, the intercomparison is an opportunity to synthesize the present state of the modeling field and range of output from various published models. This, I believe, has merit and is worthy of publication.

The topic is certainly of high relevance, but I believe the manuscript requires major revisions before it can be considered for acceptance. In its present state, the manuscript fails to present model output which is most relevant to the community. This could be alleviated by refocusing the results section with greater attention to the details of simulated infiltration between the different models.

Specific attention was given to the discussion of simulated infiltration. We decided to keep the evaluation of density and temperature because more observations are available. Additionally, appropriate simulation of these two quantities is a prerequisite to realistic simulation of water infiltration.

Additionally, I found the Discussion to be quite unorganized, reading largely as a scattered list of model details, and introducing new results. Reorganizing the Discussion, detailing how different infiltration schemes lead to wide differences in modeled density and temperature, and providing better synthesis of the model results with the objective of outlining directions for improvement, would substantially improve the manuscript.

We hope the new structure clarifies the manuscript as recommended.

These primary concerns are detailed below. In addition, I have included specific comments where language, content, and figure edits would improve the manuscript.

Major concerns
Presentation of results
As the title of the paper states, the manuscript presents a comparison of firn models with explicit schemes for accommodating the infiltration, refreezing, and runoff of surface melt. This distinguishes the study from past firn model intercomparisons (e.g. Lundin et al (2017)), and compels the authors to present intercomparison results that leverage the unique capabilities of the models. Specifically, I believe the manuscript would be of much greater interest to the community if the authors present meltwater infiltration and refreezing results explicitly. In its current form, the intercomparison results are focused predominantly on density and temperature. Minor focus is given to infiltration, refreezing, and retention, and the reader is forced to interpret the differences in infiltration/refreezing among the models through the density and temperature results.

We hope that the new structure gives more appropriate space to the comparison of meltwater infiltration. We brought to the main text several plots for the summer 2016 at Dye-2, so that it could be discussed along with percolation depth and upGPR observations. In the Discussion Section and in the Summary remarks, we also now make clear that the spread among models and the deviation from observation originate for a great part from different meltwater infiltration patterns.

The depth of infiltration at the different sites/years, and the annual depth distribution of refreezing are quantities that would be very useful in presenting the range of simulated meltwater infiltration among the models. I recognize there are few observational validation metrics, but the authors do have the upward-looking radar results (I found this figure to be the most informative in the manuscript). But even so, one of the objectives of the manuscript I believe is to communicate the range of meltwater infiltration in the retention models that are out in the community. Since infiltration/refreezing is also of first order importance in controlling the simulated density and temperature range, a greater focus on these quantities provides a much richer context for discussing the density and temperature results. This is especially true considering that observations of these variables are more common.

In figure 4, 6, 8 and 10 (in the revised manuscript) we believe we display the variety of infiltration depth and liquid water content distribution in each model. We believe that plots of maximum percolation depth at sites where no measurement of that quantity is available, would not have added information compared to the plots of liquid water content that are already in the manuscript. Nevertheless we did our best to describe, in the text, the model spread in maximum infiltration depth.

Organization of discussion
Much of the discussion reads as a scattered list of interesting things the authors noticed in their results, rather than a synthesis of the model results with direction for the community. This is especially true of section 5.1. I think the authors have an opportunity to use detailed infiltration results (see above) as a foundation to discuss why density and temperature are so variable. The Summit density results show that

the underlying densification schemes in the models are similar, even when temperatures span a wide range. So the large range in density results at percolation zone sites is due primarily to differences in infiltration. This point would be strengthened if the authors present the refreezing results. By framing the density/temperature output within the context of infiltration, and comparing these (density/temperature) fields to observations, the authors can expand on how the general infiltration approaches perform and comment on directions for improvement. This approach seems more logical than the scattered listing of 10 details and improvements. It may also feed more naturally in to the discussion of density and temperature uncertainty in models.

The discussion section was remodelled site-wise to describe what we learn about the firn models in each firn area represented by our sites. We hope that this new organisation improves readability and highlights the model spread in infiltration. The lessons learned at each site and for each model are now also summarized in the Summary remarks.

Section 5.2 focuses on uncertainty in density, temperature, and mass balance from the suite of model output. I found the mass balance uncertainty particularly interesting, but I also consider these to be results, and should be presented as such in section 4 (Results).

We have now developed the description of inter-model variability in calculated retention and runoff. We agree that in that part we present new material and it is therefore not suited in the Discussion section. However, we believe that this novel uncertainty estimation is more a development to the results that have been presented and discussed in the rest of the manuscript. We have therefore dedicated a section, entitled "Towards ensemble-based uncertainty estimates for firn model outputs", to this work.

The Discussion section here is a unique opportunity to discuss the implications of the infiltration schemes for quantities that are of perhaps the most importance to many -- runoff. The authors offer discussion of specific sites but any sort of upscaling is lacking.

We now discuss the inter-model variability in meltwater retention and runoff calculations. Since this new part builds on top of the results presented and discussed in the rest of the manuscript, it can be found in Section 6: "Towards ensemble-based uncertainty estimates for firn model outputs".

Moreover, the results from Dye-2 are used to predict future conditions at the site, which I believe is both a non-sequitor and not necessarily supported by the results.

Agreed, we removed this statement.

Based on the presented results from KAN_U it is apparent that the different infiltration schemes result in a huge range in runoff/refreezing from model to model. If the Machguth (2016) results are to be accepted, then it suggests that perhaps the retention models underestimate runoff. However, Machguth et al. (2016) also define Dye-2 as a deep percolation site outside the runoff regime. Yet, the DTU model, which shows the highest runoff at KAN_U, also indicates that 30-40% of surface melt runs off at Dye-2. Based on: a) the wide spread in modeled runoff, and b) the fact that the model which supposedly most closely honors the interpreted runoff at KAN_U also appears to overpredict runoff at Dye-2, my interpretation is that the

current generation of infiltration/refreezing models do not have the capacity to quantify runoff with fidelity. Is that an accurate assessment? If so, this would be a very important point to make to the community.

Indeed. We agree that there is currently no model that simulates infiltration and refreezing correctly at all sites. We now highlight it in the abstract and conclusion.

I struggled with section 5.3. It outlines many shortcomings of the models' ability to treat firn aquifers but does not draw explicitly from anything in the Results. The Results section simply states that the case of the firn aquifer is discussed in 5.3 (line 312). If the authors are to discuss model shortcomings, then they need to at least support these criticisms with presented results. Restructuring the results section by site (presenting infiltration, density and temperature site by site) would be one way to help achieve this. This way, for instance, the reader will be primed on whether models simulate firn reaching the melting point down to 20m as is shown in observations. This appears to occur in a number of models in Figure 5 but the text in the Discussion implies that this may not be the case.

The new site-wise organisation and better description of the result at the FA site should have clarified these points.

Specific comments
ln 28: change '...capacity to retain part of the surface meltwater...' to '...capacity to retain part or all of the surface meltwater...'
Revised for "has the capacity to retain surface meltwater"

Abstract: The abstract contains only methods and results. No synthesis of interpretations or summary statement.

The abstract was revised to provide better synthesis of the results.

ln 44: Consider replacing 'seen' here and elsewhere in the manuscript. 'Occurred'?

Rephrased to "Increased surface melt in the firn area of the Greenland ice sheet affects firn structure…"

ln 45: See comment re: ln 44
Revised to "seldom observed"

ln 60-65: In the first para, the importance of firn for meltwater retention and observed changes are established. In this second paragraph it would be stronger to communicate the challenges associated with meltwater infiltration in firn. Reijmer presented early work modeling infiltration/refreezing. But new models are on line with differing and, in some instances, more physically based schemes for meltwater infiltration/refreezing. Others have also tested different melt infiltration schemes. What about Steger (2017)? The authors need to do a better job of placing this work in the context of existing comparisons of meltwater infiltration/retention.

We added: "Steger et al. (2017) and more recently Verjans et al. (2019) investigated the impact of meltwater infiltration schemes on the simulated properties of the firn in Greenland. These studies highlighted the potential of deep-percolation schemes, for instance for the simulation of firn aquifer, but also the sensitivity of simulated infiltration to the firn structure and hydraulic properties. In these previous studies, the surface conditions were prescribed by a regional climate model. Inaccuracies in this forcing could therefore explain some of the deviation between model outputs and firn observations and prevented a full assessment of different firn model designs."

ln 66: Why just 'some' of the models currently used on the GrIS? What other models are the authors explicitly not using, and why is this the case?
Only models that replied positively to the openly advertised request to participate in the experiment are included in the comparison. The nine participating models are among the most commonly used firn models. We have reformulated the sentence to indicate that a set of nine models participated in the experiment.

ln 70-73: I would consider reframing the questions to more explicitly treat the different infiltration schemes. Isn't this the point of the study, to compare the infiltration schemes of different model schemes and the impact on the firn framework (density and temperature), and resulting runoff/refreezing partitioning?
We revised these science questions to a more straightforward description of the study's structure:
At each site, we compare simulated temperature, density and the resulting meltwater infiltration patterns between models and to in situ measurements. We discuss model features that can be responsible for model spread and deviation from observations. Lastly, we evaluate how differences in simulated firn characteristics result in various simulated refreezing and runoff values at sites where melt and/or runoff occur and attempt to quantify uncertainties linked to firn models.

ln 117-119: Reeh (2005) limits melt to the annual layer while the bucket approach redistributes based on cold and liquid water content. Seems to be contradictory?
Thank you for bringing this to our attention. We removed the reference to Reeh (2005) which was inappropriate.

ln 124-125: Is this statement relevant to the output? It appears to be a technical detail of the model mechanics that is a non-sequitor.

Indeed, this sentence was not necessary to the understanding of the main model characteristics. It was removed.

ln 141: Vandecrux (in review) does not appear in the references?
Reference updated

Table 2: GEUS model, Runoff Calculation -- change 'identical' to 'adjacent'? Also, DMIHH Meltwater routing is empty. Should this also be Darcy's law?

Revised to "adjacent" and added Darcy's law to Table 2, thank you.

ln 210: The authors state that 'as many boundary conditions as possible' were given to all modes. Is there any guarantee that the models are all being forced by the same surface melt? This is imperative to understanding the results. Hopefully the answer is 'yes'. I believe, based on reading of the supplemental, that this the case for all but the Meyer and Hewitt model? I think this should be stated explicitly in the manuscript. Moreover, the authors should state in the main manuscript an estimate of the high/low melt bias in the Meyer and Hewitt model.

We stated this more clearly:

This surface energy and mass balance provides, at three-hourly resolution, the three surface forcing fields that were used by all models: the surface "skin" temperature, the amount of meltwater generated at the surface, and net snow accumulation (precipitation – sublimation + deposition). Only the MeyerHewitt model required minor adaptation of these forcing fields (see Supplementary Text S1).

Figure 4: With such narrow panels this plot is impossible to decipher. Are models grouped and cores repeated 3x for clarity? What about 2 plots for each core? 1) the measured core density, and 2) the difference between measured and modeled for each model?

This plot was updated. All curves are in the same panel now.

Figure 6: Again, the plot is indecipherable. All the text overlaying the plots implies that the visuals themselves are unimportant. If so, then just present the ME, RMSE, and R2 in a table and be done with it.

We understand that it is a lot of information on the same plot. However we believe that the colored background still allows (with 100% zoom) to see if the cold/warm bias is located closer to the surface or down at depth or if there is a seasonal pattern in the model deviation. We would therefore like to continue with this format.

ln 303: 'more' is very qualitative. The results also appear to be very model-dependent at KAN-U. So what is 'more'? Consider just eliminating it.

This section has been rewritten.

ln 319-321: I found this lead-in confusing b/c the next figure call (Figure 7) references is all sites. Further, the next Figure 8, is focused on Dye. As stated above, I would consider restructuring the results by site.

Reorganized as suggested, to discuss one site at a time.

Figure 8: State in the caption that a, b, c are all the same. Just grouping models for clarity.

All curves are now presented in one panel.

ln 339-340: awkward sentence
The sentence was revised.

ln 369: what is 'dissipate contrast'?
The sentence was revised.

ln 382: What is inappropriate? Nearly all of the models use a thermal conductivity referenced in the literature.
Revised to "inaccurate estimates of".

ln 384: Incomplete sentence.
Revised, thank you.

ln 407-409: Interestingly, Charlampidis et al. (2015) show what they believe to be refreezing below this very same ice slab. Just delayed in time.

Thank you for pointing this out. We changed the wording accordingly:
We note that the ice slab has a low, but not null, permeability as illustrated by rarely observed meltwater refreezing events within the ice slab (Charalampidis et al., 2016).

ln 414-415: This could then indicate the challenge of comparing a firn model, which operates over such large spatial scales, with measurements of density that are local and prone to large spatial variability.

We believe this is already covered in this same paragraph. We specify what are the scale-related assumptions used by each model and conclude that "We recommend further investigation of the permeability of ice-dominated firn in relation to the firn density, the ice layer thickness and the various spatial and temporal scales at which the firn models are used.". Unfortunately, we do not have data here to assess these scale-related assumptions.

ln 526-527: These are new results in the conclusion!

These results are now stated more clearly in the result section.

ln 532: The paragraph leading to this statement summarizes what I believe to be the primary purpose of the manuscript. Synthesis of the model results beyond site-by-site agreement would vastly improve the paper. The final sentence is also incomplete.

This sentence of the conclusion was meant to give perspective for future research (new observations of horizontal/vertical water flow and spatial representativity of firn models). We completely agree that a study addressing these questions would have great impact. Unfortunately, we do not have the data, nor the model runs to investigate these. We nevertheless want our study to be a steppingstone to the next leap in firn modelling, identifying modelling strategies that are improper to describe certain firn structures or model parameters that are currently completely unconstrained. We hope the new structure makes our objectives more clear.

Supplemental: Considering the revisions required in the manuscript, I have not gone through the supplemental material in detail. However, figure call-outs do not follow a consistent order (e.g. Figure ST1 is followed by Figure S1).

Thank you, this issue has been fixed.

---

## Author Comment (AC2) · 10 Jul 2020

Dear Dr. FitzGerald,

Thank you for reading and commenting on our study.
Please see our response in green below.

Sincerely,
Baptiste Vandecrux on behalf of the co-author
* * *
This report is significant in that it compares the performance of multiple firn models in Greenland, run using the same forcing data and boundary conditions, to field observations. The importance of this work, however, is obscured by the poor organization of the report, particularly in the discussion section. Improvements to the organization of the report through the implementation of an interpretation framework and systematic discussion would increase its relevance to the modeling community.

The discussion currently reads as a list of findings, instead of taking the reader through the results in a systematic way. Organizing the discussion using a framework (e.g. in terms of model type, site, result type) similar to that used in the results section would improve clarity. Additionally, it is unclear why some sections are included in the discussion section. For example, section 5.1.10 discusses the importance of the value used for fresh snow density in the model; however, this section cites only previous studies and does not discuss how this is demonstrated by the results of this work.

As suggested, the Discussion Section has been reorganized to cover each of the four sites/climate zones: dry firn, percolation, ice slab and aquifer regions.

Additionally, with the many abbreviations used for model and site names, keeping track of the properties of each model without constant reference to Table 2 is difficult if the reader is not familiar with all the models. Having a framework in which these models are referenced, and a more systematic discussion of results, would improve readability.

We understand the low readability of the model name. Unfortunately it was not possible to change them at this point. Nevertheless, we now always mention the characteristic of the model that we want to discuss, along with its name: e.g. *"only the models with explicit deep-percolation schemes (CFM-Cr, CFM-KM and UppsalaUniDeepPerc) simulate water below 10 m depth"*.

This report is of particular interest to those deciding between firn models to use and those interpreting results from such models. As such, the results of this paper could be presented in a way to highlight the effect that these findings have on choosing or interpreting these models. These findings are touched on in the abstract but are difficult to parse from the discussion. Better organization of the discussion, as previously suggested, would remedy this issue.

In addition to the restructuration of the Results and Discussion Sections we now also summarize and highlight the strengths and weaknesses of different models in the Summary remarks and perspectives Section.

---

## Author Comment (AC3) · 10 Jul 2020

Dear Dr. Morin,

Thank you for your thorough review of our manuscript and for your constructive suggestions. Please see below our response to each of your remarks.

Sincerely,
Baptiste Vandecrux on behalf of the co-authors

The manuscript by Vandecrux et al., entitled "The firn meltwater Retention Model Intercomparison Project (RetMIP): Evaluation of nine firn models at four weather station sites on the Greenland ice sheet", provides results from a recent intercomparison of firn models dedicated to their handling of meltwater retention and runoff. This topic has attracted a lot of interest over the past decade, following the discovery of subsurface aquifers on the Greenland ice sheet and the fact, more generally, that liquid water transfer is a critical, yet very complex process governing the energy and mass balance of firn below the surface. The manuscript undoubtedly fits the scope of The Cryosphere. It is generally well designed and easy to understand, although, like many other manuscripts reporting on intercomparisons, the reading can be a bit cumbersome and the results will mostly appeal to experts in this field. I recommend publication of this article, and I provide here below some feedback and suggestions, which may be considered by the authors in case they lead to improvements of the manuscript.

Page 1, line 1-2: "Perennial snow, or firn, covers 80% of the Greenland ice sheet and has the capacity to retain part of the surface meltwater, buffering the ice sheet's contribution to sea level". This sentence could in fact be quite misleading and I suggest reformulating, either at least acknowledging that the "buffering" acts upon sea level *rise*, or maybe more generally, leave out the term "buffering" (which could be inappropriate in some cases where runoff generation can increase the contribution of the ice sheet to sea level rise), and use a more neutral term referring to the fact that firn processes influence the behaviour of the ice sheet and affect its contribution to sea level rise.

Revised as suggested: Perennial snow, or firn, covers 80% of the Greenland ice sheet and has the capacity to retain surface meltwater, influencing the mass balance of the ice sheet and its contribution to sea level rise.

Page 1, line 3 : "weather-station-derived" : maybe consider unpacking the wording, this is quite tedious to read.

Revised as suggested: forced by mass and energy fluxes derived from weather stations at four sites.

Page 2, line 60 : I think it could be useful, and appeal to a wider community of readers, if this article could provide more information on what a "firn" model is, or what a "snow and firn" model is if the two terms here are meant to be combined. Indeed, GCMs and RCMs all feature a "snow" component in their land surface models, and many such models do not handle ice sheets differently from a long-lasting seasonal snowpack. It could thus be quite appropriate to provide more background on how snow and firn processes are considered in GCMs and RCMs (or other tools used for reanalyses etc.), in order to make this effort even more useful in a CMIP6/IPCC context. This could make it possible to establish how the firn models used for this intercomparison fit in this wider context, thereby providing information on

whether the results of this intercomparison are relevant when discussing the results of existing GCMs and RCMs projection. Also(see below point too), it would be good to introduce what are the typical input/output to firn models, because they seem to deviate from typical input/output of land surface models, hence the link to GCMs/RCMs/etc. is not direct.

Regarding the use of firn model in GCM/RCM we added: "The performance of these models, when coupled to regional and global climate models, has a direct impact on the quality of ice-sheet mass-balance calculations (Fettweis et al., 2020) and sea-level change estimations (Nowicki et al., 2016).". A general description of the firn model structures and traditional use is presented in the Method section. Since it is rather technical, we did not want to have it in the Introduction.

Page 2, line 64 : Although I know this can be debated, I have strong personal reservations against the term "validation", which I believe is beyond reach in any geosciences field because the "truth" is never known hence "validation" (in the strong sense) can only be elusive. I much prefer the term "evaluation", which better encapsulates the fact that the evaluation results from a comparison with observations, which also carry some uncertainties. I was happy to see the term "evaluation" in the title and in the abstract, so it was disappointing to see the term "validation" popping up in the introduction. Maybe this can be harmonized throughout. Along the same line, in several instances the comparison between model output and observations is referred as a "bias". Here again, "bias" is a term which includes some judgement of value, implicitly assuming that observations are the "truth". I think that observations are never the "truth" and, especially regarding in-situ snow and firn observations, we know that observations are intrinsically prone to significant errors, in addition to large spatial variability for many variables at all scales, which induces representativeness issues. In this context, I much prefer referring to "deviations" between model results and observations, without using the term bias. I note that the term "deviation" is used in several places in the manuscript, including in figure captions (e.g. caption of Figure 6, which I think is perfectly worded), so maybe this could be harmonized in the text.

We agree with these points and have revised the text to harmonize the use of the terms "evaluate" and "deviations", recognizing that true validation is a difficult exercise, especially in field settings around variables that vary widely in space and from year to year. That said, some model results are arguably invalid or biased (e.g., deep infiltration of meltwater and anomalously warm firn temperatures at Summit, counter to available observations and physical expectations), so we still use of the term "bias" in the manuscript.

Page 2, line 67 : Here is a good example where more information could be provided on whether (some of) the firn models included in this intercomparison are representative to how firn processes are handled in GCM/RCM/NWP models, so that the results can be used to analyse some of the output of such models. At present, and even though some of the information is provided in section 2, the models included in the intercomparison are not categorized explicitly according to their use, and I think it could be helpful to the scientific community to provide the rationale and the results of the intercomparison in a way that can (somehow) be transferred to the interpretation of other model results.

Considering that the history and description of each model are already reported in the Methods section, we would like to keep the introduction concise and not list the models there (which would require

references and explanation of the acronyms). We also hope that the introduction now connects better the RetMIP with broader applications of firn models.

Page 3, line 91 : it seems that there is a typo in this reference, I believe this should be 1998. I haven't checked all reference, but if they were typed by hand and not using literature management software, there may be other errors in the references.

Revised to 1998, thank you.

Page 4, line 1 : This table is very useful, I suggest adding a column for providing the extended name (developing the acronym) and, more to the point, adding a column on how the model is typically used (included/coupled to a land surface model in RCM/GCM/NPW context, or purely offline for process investigations etc.). I'm convinced that the authors can easily define several categories within which models can be classified (these categories could even be used in the results and discussion, if common features, or not, emerge from these various categories, in addition to discussing results referring to process representation in models).

We thank you for this suggestion but would like to keep the model use within each model section within the method and not as a grouping criterion. Many of the models have been used for various purpose but with very different settings, making it unclear whether a label apply to the model set used here. Lastly, I would like to point out that, like for other earth surface models, the choice of a firn model for a specific application is rather determined by "legacy rather than adequacy" (https://doi.org/10.1029/2018WR022958). With our study, we hope we highlight the tasks for which each model is adequate independently of how it has been used in the past.

Page 4, Line 98 : I think it would be good to spell out the acronyms in the titles of subsections 2.1, 2.2 etc., this is otherwise quite obscure for non-expert readers not accustomed to the acronyms of these firn models.

To keep the titles of subsections concise we wish to keep the acronyms there. They are then defined in the paragraphs just below. Another motivation for this choice is that most of the acronyms used as model names do not carry information about the model type or design and therefor are not deemed worth to be highlighted in the subsections' titles. We note that Fettweis et al., (2020, https://doi.org/10.5194/tc-2019-321) used a similar strategy.

Page 5, line 116 : extra "-" between ASIRAS and instrument

This sentence was revised.

Page 5, line 131 : it seems to me that layers defined by a w.e. (e.g. mass per unit surface area) should not be qualified by a "thickness", which refers to a distance (in m). If the model is formulated in terms of layers with a given mass, then I believe the text should refer to this, and the substitution of "thickness" with "mass" will accurately represent how the model is formulated.

We agree that the phrasing was awkward here. We changed for: " DMIHH employs 32 layers within which snow, ice and liquid water fractions can vary and where each layer has a constant mass."

Page 7, Table 2 : I think that the column on "Hydraulic conductivity" needs some attention. The van Genuchten (1980) article provides a way to link between the saturated hydraulic conductivity and the hydraulic conductivity, which for snow has been addressed by several studies such as Shimizu (1970), Calonne et al. (2014) or can be used using geometrical estimates such as Carman-Kozeny (see Calonne et al., 2014 for a review of existing formulations, and Wever et al., 2014, for context). Hence I suggest to double check, for each model, what is the parameterization used for estimating the saturated hydraulic conductivity (corresponding to the permeability) from the microstructure (density, specific surface area/grain size) and the formulation used to derive the hydraulic conductivity (van Genuchten (1980) is probably widely used). This column seems to be lumping and mixing the two.

Thank you for spotting this. We now specify for each study using Darcy flow  the saturated and unsaturated hydraulic conductivity, with the source of the coefficients that have been used if necessary.

Page 8, line 177 : I'm not fully convinced by the formulation of how the forcing data are introduced. "Any bias in forcing data propagates into the model output" : I'd rather suggest that any *difference* in forcing data propagates into differences in model outputs. The term "bias" is here inappropriate I think (see comments above).

Here we respectfully disagree and would like to continue with the word "bias". Regional climate models are known to have systematic deviation from observations both locally (AWS) and on a larger scale (against remote sensing products). For example, Noël et al. (2018, https://doi.org/10.5194/tc-12-811-2018) use 93 times the word bias when describing RACMO2.3p2.

Further, "To make sure we compare and evaluate the models independently of biases that may exist in forcing datasets that come from RCMs, we use meteorological fields derived from five weather stations at four sites." I think this sentence needs rephrasing, because it gives the impression that only RCM atmospheric fields can be biases, and in-situ atmospheric observations are not biased. I don't see the point in referred to RCM here at all, but simply state that "To make sure we compare and evaluate the models independently of differences due to forcing data, we use for all models the same meteorological fields derived from five weather stations at four sites." The references to RCM data is absolutely not needed here, and in the current formulations I consider it misleading and improper.

We agree that AWS data can be biased and we mention weaknesses in our dataset (f.e. lack of tilt correction for the radiation data) to be corrected in future intercomparisons. But we believe that AWS are the best estimation of local meteorological conditions. We would like to continue with the phrasing and raise awareness about the deviations that exist between RCM and AWS measurements: for instance RACMO2.3p2 (Noël et al., 2018) give air temperatures that are on average 2.7°C colder than observed at Summit station.

This is now also introduced in the introduction when mentioning previous model intercomparisons: "Steger et al. (2017) and more recently Verjans et al. (2019) investigated the impact of meltwater infiltration schemes on the simulated properties of the firn in Greenland. These studies highlighted the potential of deep-percolation schemes, for instance for the simulation of firn aquifer, but also the sensitivity of simulated infiltration to the firn structure and hydraulic properties. In these previous studies, the surface conditions were prescribed by a regional climate model. Inaccuracies in this forcing could therefore explain some of the deviation between model outputs and firn observations and prevented a full assessment of different firn model designs."

Page 9, line 202 : If I understand well, the firn models are driven by 3-hourly skin temperature, meltwater generation (what it this ?) and net snow accumulation.

Indeed, we clarified the phrasing: "This surface energy and mass balance provides, at three-hourly resolution, the three surface forcing fields that were used by all models: the surface "skin" temperature, the amount of meltwater generated at the surface, and net snow accumulation (precipitation – sublimation + deposition). "

I think this warrants an explicit statement on the forcing data for firn models (see my comment regarding the introduction), because this appears to be quite different from forcing data of land surface models (including the snow component), usually driven by air temperature, relative humidity, incoming shortwave radiation, incoming longwave radiation, wind speed (and direction) and snowfall and rainfall rate (in offline or online applications).

I do not have much experience with reanalysis datasets and land surface models but RCMs like HIRHAM and RACMO use the same energy budget closure approach to calculate surface temperature and surface melt that are then passed to the firn module:

*At the surface, snow mass is updated with snowfall, rainfall, melt and deposition/sublimation at each subsurface scheme time step (1 h). Likewise, the surface temperature is updated via energy budget closure with radiative and turbulent surface energy exchange above and diffusive and advective heat exchange with subsurface layers. If the surface temperature exceeds 0°C, it is reset to 0°C and the excess energy supplies heat for melting (Langen et al., 2015).*
Langen et al. (2017, https://doi.org/10.3389/feart.2016.00110)

*In RACMO2, the skin temperature (Tskin) of snow and ice is derived by closing the surface energy budget (SEB), using the linearised dependencies of all fluxes to Tskin and further assuming, as a first approximate, that no melt occurs at the surface (M D 0). If the obtained Tskin exceeds the melting point, Tskin is set to 0 C; all fluxes are then recalculated and the melt energy flux (M > 0) is estimated by closing the SEB."*
Noël et al. (2018, https://doi.org/10.5194/tc-12-811-2018)

The forcing fields that were provided to the RetMIP participants were therefore close to what each of these firn modules traditionally take as input. Surface models that could not take these prescribed forcings, and required to be given meteorological fields instead, were not considered in our study as they

would have had different temperature and meltwater input at the top of their simulated firn column. We do not see the need for listing the firn models that are not compatible with our forcing fields. The MeyerHewitt model was an exception to this since Colin Meyer found a meaningful way to re-calculate the surface energy fluxes that would give, for his energy balance scheme, similar surface temperature and melt as prescribed (details in the supplementary material).

In this context, it would be good to quickly introduce how the firn model data input are typically computed within GCM/RCM/NWP models where they are implemented.

In the introduction we now mention "Firn models traditionally take as input energy and mass fluxes at the surface and calculate the evolution of firn characteristics and meltwater retention at scales ranging from tens of metres to tens of kilometres ". Making a review of currently-used surface energy and mass balance models that can be used to force firn models (from full energy budget to PDD and statistical approaches), is out of the scope of this manuscript.

I also think that it would be beneficial for the manuscript, if the material provided in the Supplement regarding how the models were adapted in order to contribute to the intercomparison, could be placed within the body of the article. [In fact, I'm in favour of moving the whole supplementary material into the article, which is quite technical anyway, and which I think would benefit from having all the information in the same document].

We brought in the main text Table S1 (now Table 4) and removed some of the unnecessary material from the supplementary. Considering the length and the complexity of the study we would like to keep the remaining information (complementary information regarding our methods ) in the supplementary and only have the necessary information in the main text. If there is a specific result or discussion point that cannot be understood without the supplementary, we will be happy to move the required material to the main text.

Page 10, Table 3 : Mean annual air temperature data should probably be homogenized in terms of formatting (why is the number provided by 0.1C resolution of KAN_U and rounded to the unit C for other sites ?)

Revised to common precision (nearest degree).

Page 20, line 371 : Eulerian

Revised.

Page 20, line 384 : it seems that the sentence does not end as planned. Was it the intent to refer to Calonne et al. (2019 ; https://doi.org/10.1029/2019GL085228 ) which may provide some hints into how to parameterize firn thermal conductivity ?
The sentence fragment has been corrected and this reference has been added - thanks for flagging this very relevant paper.

Page 21, line 404 : The last sentence "Especially, the heterogeneous nature of the firn, the presence of vertical ice features in the firn, the variability in surface snow density/thermal conductivity as well as firn ventilation are processes not currently included in the models and should be subject of future research." is certainly true but is quite vague. Based on existing literature (e.g. Albert et al.) is it possible to elaborate more on the expected direction of change and whether accounting for such processes would be beneficial (and at what numerical cost) ?

Following the recommendation from Reviewer 1, we now only elaborate on topics for which we have data and/or model outputs. Topics such as spatial heterogeneity, firn ventilation or surface snow density are only mentioned for future research and are therefore outside of the scope of this study.

Page 21, line 413 : add "with" after "match"

Revised.

Page 22, line 433 : on the "fresh snow" density issues and beyond, there is little discussion in the manuscript on the connection between snow cover models (such as those embedded in land surface schemes) and firn models. Wouldn't it be adequate that surface processes are handled by snow cover model rather than firn models, taking advantage of the features of both types of models ? This connects with the question on how firn models are used within coupled GCM/RCM or offline (see comments above).

As recommended by reviewer 1, we reduced the discussion of surface snow density as we do not present data or model output bringing new insight on this topic. We agree that surface snow density is tightly linked to the surface meteorology and consequently opens the question of how firn models are coupled to surface and atmospheric conditions. But we wish to focus here on the inter-comparison of firn models and leave this issue to future work.

Page 22, line 443 : It should be mentioned here that the uncertainty envelope corresponds to +/- 2 standard deviation (+/- 2 sigma). This information is missing here (although it is provided later in the conclusion). Regarding the temperature, I see a limitation to the estimate provided here, in the sense that the temperature of the firn cannot exceed 0C. How does this impact the uncertainty range provided for temperature?

Now clarified where we first introduce the uncertainty estimation. Indeed, as a consequence of this upper bound of firn temperature, model spread indeed decrease at sites where firn reach 0°C. It simply implies that it is easier to simulate firn temperature at a temperate site than at a percolation or dry snow site.

Page 23, line 451 : What is the argument for stating that this uncertainty range applies in situations where observations are not available ?

We have reworded this for clarity; they are modelled uncertainties, in the absence of observational constraints. Where observations are available, it is possible to calibrate models or reject models that do not perform adequately, reducing the model uncertainty envelope.

Page 24, line 482 : "much faster drainage of the aquifer" : faster than what ?

Sorry for the ambiguity: faster than the Zuo-Oerlemans parameterization. Now clarified in the text.

Page 24, line 484 : A qualifier is probably missing before "models" : "existing firn models" ? "firn models considered in this intercomparison" ?
Revised as suggested.

Page 24, line 490 : I suggest not using the term "reality" in scientific publications. Furthermore, a reference is missing to support the statement in this sentence.

This has been reworded here and at two other instances in the manuscript.

Figures : I didn't notice any major flaw in the design or content of the figures, which are appropriate to convey the results of this intercomparison.

---

## Author Response (AR2)

Dear Reviewer, Dear Editorial team,

We are grateful for the further comments and suggestions that continue to improve the manuscript. We address each point individually below.

We additionally conducted a last proof-read of the manuscript. In that process we moved the AWS description back from the supplementary material to Section 3.1, reformatted Table 3, improved readability of most figures.

Sincerely,
Baptiste on behalf of the co-authors
* * *
The revised version of 'The firn meltwater Retention Model Intercomparison Project (RetMIP): Evaluation of nine firn models at four weather station sites on the Greenland ice sheet' by Vandecrux et al., is an improvement from the original submission. The manuscript will make a meaningful contribution to the field. I do, however, provide here some comments which I believe will improve the manuscript prior to publication. Edits to a number of the figures would improve their clarity and communication of the results. I believe the authors need to clarify to the reader if (and if so, how) preservation of the initial conditions in short model runs may skew the results. I also believe the important manuscript points regarding model runoff/retention agreement could be better communicated by presentation of expanded output in a specific results section. I have detailed these and other comments below.

Line 64: consider replacing 'quality' with 'fidelity'

**Updated**

Line 75: should be 'nine firn models'.

**Thank you, updated.**

Line 182: I don't believe 'runoff' can be used as verb. Break in to two words.

**Updated**

Line 205-206: Why is 'more recent' synonymous with higher quality? Maybe more appropriate to state that you use this forcing dataset because it is co-located with the other observations you have for evaluation metrics?

**We now specify why more recent station means higher observation quality:**

***Since this station was more recently installed than the GC-Net station, it ensures better meteorological observations (levelling, absence of frost/mist on radiometers) and therefor better forcing for the models over the 2016 melting season, during which an extensive observational dataset is available for model evaluation.***

Table 3: The forcing datasets all span very different periods of time, which has a direct bearing on the results. For instance, the authors compare the firn density at depth ranges up to 20m. It will take many years to decades for the initial conditions to

be buried beyond these depths, but none of the forcing datasets are longer than ~17 years, and at the firn aquifer site (where the climate forcing only spans 8 months), the model results may be largely dictated by the initial conditions. Consequently, it's unclear how much of the presented results reflect performance of the models, and how much is simply due to the continued presence of initial conditions in the modeled temperature and density.

**Indeed the model performance depends not only on the model characteristics but also on the quality of the forcing and boundary conditions.**

**We added is section 5.1:**

*At Summit, the top of the initial firn density profile is advected to 10 m depth by the end of the simulation (Figure 2). Consequently, we here assess both the models' capacity to accommodate and transform new snow at shallow depth and how models densify the initial density profile as it is advected downward. The persistence of the initial conditions consequently influences the performance of the models but have the advantage of giving all models the same starting point as opposed to, for instance, spin-up procedures. This was deemed more suitable to intercompare the meltwater retention in different models. In spite of measurement uncertainty and firn spatial heterogeneity, the firn density and temperature measurements used to initialize and evaluate the models represent the closest estimation of actual firn characteristics. Additionally, important biases in initial firn density and temperature would lead to a visible adjustment of the simulated firn characteristics in the first months/years as the model reacts to the surface forcing. No abrupt change can be seen in the simulations (Figure 2), which gives confidence that the initial conditions were appropriate.*

As an aside, it remains unclear what density profiles the authors use as the initial condition; both Herron and Langway and measured profiles are presented in the SI.

**We updated this plot and removed the Herron and Langway model to make clear that the initial conditions are a combination of in situ measurements otherwise reported in Table 3.**

The authors should include some treatment of this issue in the manuscript. For the sites with longer forcing datasets (Summit and Dye-2), is it safe to say that the initial conditions have been pushed through the model domain? If so this should be stated (based on Figure 3 it appears this may not be the case?). At the other sites, KAN_U and FA, some statement regarding the influence of relict initial conditions needs to be made.

**We want to note that our approach is very different from a spin-up procedure, where a rough estimate of the initial conditions is being "pushed through" by multiple model iteration. We consider that "***the firn density and temperature measurements used to initialize and evaluate the models represent the closest estimation of actual firn characteristics"*** **and consequently do not need to be pushed out of the model domain. We, on the contrary, assess how models transform the observed initial conditions during the model run and given a certain surface forcing. We also identify misbehaviors within the models (ice slab building up at Dye-2, percolation through ice slab, water running off within the aquifer…) that are clearly linked to inadequate model design rather than to potential errors in the observations used for initialization and evaluation of the model.**

Line 236: This 'deep firn temperature' is presumably the firn temperature presented in Table 3, but at some sites it is in fact

not deep at all (5 m at KAN_U). Yet, in the case of KAN_U, Figure 6 shows that temperature measurements extend to 10m. So why this discrepancy?

**Thank you for spotting this. The measured depth was actually 8 m.**

**In fact, all these boundary temperatures are temporal averages over a long period (when possible) to compensate the shallow depth at which they are sometimes measured. We removed the measurement depth from Table 3 and added details about the averaging process. Additionally we changed for "bottom temperature" which is more neutral when describing this model feature.**

*Initial temperature profiles were calculated using the first reading of air temperature (as first guess of surface temperature), the first valid measurement of firn temperature, and the bottom firn temperature (Table 3). The bottom firn temperatures (Table 3), needed as lower boundary condition by some of the models, were calculated from the available firn temperature measurements. At KAN_U, the average of the deepest firn temperature, at ~8 m depth, was taken over spring 2013 – spring 2015 period. At Summit and Dye-2_long, the 10 m firn temperature was interpolated when firn temperature measurements were below 10 m depth and then averaged. For Dye-2_16 and FA, the deepest firn temperature measurement, at 9 and 25 m depth respectively, were averaged over their respective measurement periods (Table 3). Initial liquid water content at FA is calculated according to the observations from Koenig et al. (2014) which indicate pore saturation below 12.2 m depth. Some models also need long-term mean air temperature and accumulation (Table 3) which were calculated from Box (2013) and Box et al. (2013).*

How much of the deviation between measured and modeled temperatures simply results from the prescription of a boundary condition that is far from reality?

**Here we respectfully disagree with the fact that prescribed bottom firn temperatures are "far from reality". On the contrary, this is the closest observations allow us to get to the actual deep firn temperature. In the GEUS model, the heat flux at the bottom of the model can be tracked. An inadequate bottom temperature would force heat to be either added or withdrawn at the bottom of the model. This was not the case. Additionally, we think that inappropriate initial conditions would create an abrupt change in firn density and temperature in the first months/years of the simulation: strong warming in case of cold-biased initial temperature, fast densification if the firn is initialized too warm. This is not the case neither.**

Figure 2: I recognize the challenge in presenting these spatial/temporal model results, but the embedded statistics in panel (C) make it quite difficult to visually assess the results. This is similarly true for the figures at the other sites. Consider moving these statistics in to a separate table.

**We collapsed these statistics to two lines instead of three which further reduces the masking of the color plot.**

Figure 3: In panel (A), consider changing each subpanel so that y-axes span the same density range but are focused on different intervals. As presented, the panels do not communicate much. In panel (D), the results appear to mostly display the model

initial conditions and not a comparison to observation since the measured densities are limited to such shallow depth. Consider only plotting from 0-7 or 8 m

**Thank you for the suggestion. We adapted the axis range and changed also panel D to display another (more interesting) firn core from 2007.**

Line 318: I don't believe a comparison 'responds', as is stated. Consider alternate wording: 'tracks closely'?

**We rephrased to:**

*For each model, the simulated firn temperature at Dye-2 (Figure 4b) and its deviation from observations (Figure 4d) responds closely to the simulated meltwater infiltration each summer (Figure 4c).*

Line 345: Consider changing 'allows to assess..' to 'allows assessment of'.

**Updated.**

Line 346: See my earlier comment re: line 205-206. Why do these data present 'the best conditions'? Consider different wording.

**We hope that we now justify why the Dye-2_16 is a better forcing dataset than the Dye-2_long for a detailed evaluation of the models.**

Lines 392-394: Does the increased deviation with time have something to do with the fact that the signal from the initial conditions is slowly being evacuated from the domain?

**Figure 8a shows that high density layers are not advected up or down during the simulation period. Indeed KAN_U is now more of a superimposed ice site and do not build-up more firn each year. The increasing spreads in the modelled 1-10 m and 10-20 m average densities therefore only stem from different patterns in meltwater infiltration and refreezing (Figure 8c) and potentially of differences in firn compaction rates due to diverging firn temperatures (Figure 8b).**

Figure 9: In panel A, Why not extend the model results through the observed density marker in 2017? There are clearly model results during this time period, as shown in panel H.

**Panel H compared the last simulated firn density profile (31-12-2016) with a firn core measurement from April 2017. We understand that the plot was misleading, removed that observation from the figure and adjusted panel a-c to focus on the simulation period.**

Figure 10: This appears to be a repeat of KAN_U results, and not the Firn Aquifer.

**Thank you for spotting this. The right figure was put back.**

Section 6.2: This section actually honors most closely the title of the manuscript -- it presents an intercomparison of meltwater retention model results.

I would strongly consider presenting this output in the results…

**We believe that it is important to present and discuss the differences in firn density, temperature and meltwater infiltration in the nine models before we present the inter-model variability in refreezing and runoff totals. When reaching this part, the reader is aware of all the intricate processes that go into these total values. We consider Figure 11 more like a summary of all the uncertainties previously discussed rather than a result on its own.**

including a panel for the single model year of firn aquifer conditions,

**We are not willing to present and discuss the inter-model variability in runoff at the firn aquifer site when we know that in fact no instantaneous runoff occurs at that site (in the sense of water leaving the ice sheet through an efficient drainage system). We explain our choice:**

*We do not evaluate meltwater retention and runoff at FA owing to the major limitations that we highlighted in the current handling of firn aquifers in firn models. Indeed, modelled runoff, traditionally defined as excess water entering an efficient drainage system and leaving the ice sheet, does not occur at FA (Miller et al., 2018). Instead, the excess water saturates the firn and slowly moves downstream within the aquifer, that none of the models can represent.*

and include the DTU model results.

**The DTU model results are included in Figure 11 but are not included in the calculation of the inter-model standard deviation. We complemented our justification:**

*At Dye-2, the DTU model produces unrealistic runoff values (Figure 11c) because of the impermeability of near-surface ice layers blocking downward percolation and enhancing runoff. This highlights how a model designed for the dry snow area (Simonsen et al., 2013) can fail to capture meltwater retention in the percolation area. We therefore do not consider this model in our multi-model uncertainty estimation.*

This would give the reader valuable insight in to the model agreement with respect to meltwater retention over a range of climate conditions. After all, perhaps the central motivation for developing these models (as described in the Introduction) is to assess retention and runoff.

**It is indeed our motivation, and we identify key elements in the models' handling of firn density, temperature and meltwater infiltration that explain inter-model variability and deviations from observations.**

A discussion section focused on these results would still be quite useful for identifying why some models (e.g. DTU) are outliers, why the models are collectively challenged under certain climate conditions (e.g. those resulting aquifer generation),

**As mentioned above, we specify why the DTU model give unrealistic values. The model representation of firn aquifer is also duly discussed in Section 5.4. We strengthened our discussion of the model spread in refreezing and runoff at Dye-2 and KAN_U. Note that we relate high and low model spread to both climatic conditions and to the model features.**

[revised manuscript text omitted]

Line 676-677: This final sentence I think is a misguided direction. Are you setting the stage for another intercomparison project? Rather than tips for future model intercomparisons, summarize the inter-model agreement at high melt sites --> it appears to be poor.

**Indeed this sentence was misleading. We rephrased to:**

[revised manuscript text omitted]

---

## Author Response (AR3)

**Response to the editor**

Dear editorial team,

We thank our scientific editor for his suggestions, which are now implemented in the new version of the manuscript.

Please note that we uploaded high quality files for the figures as supplementary. They should be used instead of the images present in the pdf.

Sincerely,

Baptiste Vandecrux on behalf of the co-authors

---

## Author Response (AR4)

**Response to the editor**

Dear editorial team,

We thank our scientific editor for his suggestions, which are now implemented in the new version of the manuscript.

Sincerely,

Baptiste Vandecrux on behalf of the co-authors